# Atrial Fibrillation and Reperfusion Therapy in Acute Ischaemic Stroke Patients: Prevalence and Outcomes—A Comprehensive Systematic Review and Meta-Analysis

**Jay Patel** [1,2,3] **and Sonu M. M. Bhaskar** [1,3,4,5,*,†]

1. Global Health Neurology Lab., Sydney, NSW 2150, Australia
2. South Western Sydney Clinical Campuses, University of New South Wales (UNSW) Medicine and Health, UNSW Sydney, Sydney, NSW 2170, Australia
3. Neurovascular Imaging Laboratory, Clinical Sciences Stream, Ingham Institute for Applied Medical Research, Sydney, NSW 2170, Australia
4. NSW Brain Clot Bank, NSW Health Pathology, Sydney, NSW 2170, Australia
5. Department of Neurology & Neurophysiology, Liverpool Hospital & South Western Sydney Local Health District (SWSLHD), Sydney, NSW 2170, Australia
* Correspondence: sonu.bhaskar@globalhealthneurolab.org or bhaskar.sonu@ncvc.go.jp; Tel.: +81-90-9274-1265
† Current address: National Cerebral and Cardiovascular Center (NCVC), Department of Neurology, 6-1 Kishibeshimmachi, Suita 564-8565, Osaka, Japan.

**Abstract:** Atrial fibrillation (AF) significantly contributes to acute ischaemic stroke (AIS), yet its precise influence on clinical outcomes post-intravenous thrombolysis (IVT) and post-endovascular thrombectomy (EVT) has remained elusive. Furthermore, the overall prevalence of AF in AIS patients undergoing reperfusion therapy has not been clearly determined. Employing random-effects meta-analyses, this research aimed to estimate the pooled prevalence of AF among AIS patients undergoing reperfusion therapy, while also examining the association between AF and clinical outcomes such as functional outcomes, symptomatic intracerebral haemorrhage (sICH) and mortality. Studies comparing AF and non-AF patient groups undergoing reperfusion therapy were identified and included following an extensive database search. Forty-nine studies (n = 66,887) were included. Among IVT patients, the prevalence of AF was 31% (Effect Size [ES] 0.31 [95%CI 0.28–0.35], $p < 0.01$), while in EVT patients, it reached 42% (ES 0.42 [95%CI 0.38–0.46], $p < 0.01$), and in bridging therapy (BT) patients, it stood at 36% (ES 0.36 [95%CI 0.28–0.43], $p < 0.01$). AF was associated with significantly lower odds of favourable 90-day functional outcomes post IVT (Odds Ratio [OR] 0.512 [95%CI 0.376–0.696], $p < 0.001$), but not post EVT (OR 0.826 [95%CI 0.651–1.049], $p = 0.117$). Our comprehensive meta-analysis highlights the varying prevalence of AF among different reperfusion therapies and its differential impact on patient outcomes. The highest pooled prevalence of AF was observed in EVT patients, followed by BT and IVT patients. Interestingly, our analysis revealed that AF was significantly associated with poorer clinical outcomes following IVT. Such an association was not observed following EVT.

**Keywords:** atrial fibrillation; stroke; prevalence; outcomes; reperfusion therapy; thrombolysis; thrombectomy

## 1. Introduction

Atrial fibrillation (AF) is a cardiac arrythmia that holds significant global importance due to its high prevalence, clinical implications, and potentially serious complications [1]. It stands as the most common arrythmia [2], with a prevalence estimated at 0.51%, escalating to 10–17% in individuals aged 80 and above [3]. The burden on public health systems and resources is substantial [1,4,5], primarily due to complications such as acute ischaemic stroke (AIS) [6], a consequence of the stasis-induced thrombus formation within fibrillating atria [7]. These thrombi may embolise to cerebral circulation, leading to AIS [8]. Although

approximately 23.7% of AIS or transient ischaemic attack (TIA) patients have underlying AF, AF frequently remains undiagnosed due to limitations of current cardiac monitoring methods [9]. Diagnosis of AF can enable interventions such as prophylactic anticoagulant therapy, often utilizing agents like Vitamin K Antagonists (VKAs) or Non-Vitamin K Oral Anticoagulants (NOACs), potentially preventing a considerable proportion of strokes [10,11]. Notably, the chronic administration of oral anticoagulants has been shown to significantly reduce the risk of ischaemic stroke by up to 64% [12]. Despite the consistent use of these agents or the maintenance of an appropriate International Normalised Ratio (INR), individuals with AF remain susceptible to the potential severity of ischaemic stroke due to the presence of concurrent factors that frequently accompany AF [13–16]. It is important to acknowledge that the efficacy of these medications lies primarily in preventing embolic events, and they may not comprehensively address a broader spectrum of associated risk factors [16]. Following AIS, patients might undergo reperfusion therapy through intravenous thrombolysis (IVT) or endovascular thrombectomy (EVT) [17]. EVT aims to mechanically remove thrombi [18], whilst IVT strives to dissolve thrombi by cleaving their fibrin network [19,20]. Eligible patients could receive a combination of IVT and EVT, termed bridging therapy (BT) [21].

Reperfusion therapy has become standard in the acute management of ischaemic stroke patients [17]. While various studies have reported on AF prevalence among AIS patients undergoing IVT [22–24], EVT [25–27] and BT [28–30], a comprehensive estimation of the pooled AF prevalence via meta-analysis is still lacking. Comparative analysis of AF prevalence between IVT, EVT and BT will additionally reveal which form of reperfusion therapy is most likely to contain AF patients. Examining regional variations in AF prevalence may further suggest areas where underdiagnosis is more likely [31]. This data is critical to develop evidence-based policy to guide resource allocation for AF screening, stroke prevention and lifestyle-related interventions [5,32]. Despite the use of reperfusion therapy, it is also not clear how AF impacts the effectiveness of these treatments [33,34]. Resolving this uncertainty holds paramount importance in devising optimal treatment strategies, as well as for risk stratification and prognosis communication with patients [35].

This meta-analysis seeks to assess the pooled prevalence of AF in AIS patients receiving reperfusion therapy and the impact of AF on clinical outcomes subsequent to reperfusion therapy. The clinical outcomes encompass 90-day functional status, symptomatic intracerebral haemorrhage (sICH), and 90-day mortality. To address these aspects, our study aims to answer the underlying key questions:

1.  What is the prevalence of AF among AIS patients treated with each type of reperfusion therapy?
2.  Is there an association between AF and a favourable 90-day functional outcome in AIS patients treated with each type of reperfusion therapy?
3.  Does AF correlate with the occurrence of sICH in AIS patients treated with each type of reperfusion therapy?
4.  Is AF associated with 90-day mortality in AIS patients treated with each type of reperfusion therapy?

## 2. Methods

### 2.1. Literature Search: Study Identification and Selection

Studies were retrieved from the databases of PubMed, Embase and Cochrane. This study was conducted in accordance with the Preferred Reporting Items for Systematic Reviews and Meta-Analyses (PRISMA) guidelines and the Meta-analysis of Observational Studies in Epidemiology (MOOSE) checklist (Supplemental Tables S1 and S2). The PRISMA flowchart depicts the number of studies identified, screened, and included in the study (Figure 1). The search was executed on 17 April 2023, with no restrictions on the date of publication. However, our final analysis encompassed IVT studies published from 2005 onwards, and EVT and BT studies published from 2015 onwards. Keywords in the search included "atrial fibrillation", "ischaemic stroke", "reperfusion therapy", "thrombolysis", "thrombectomy" and "bridging thrombolysis". Medical Subject Headings (MeSH) terms

were used in PubMed to refine the search [36]. Filters were applied to exclude non-English language studies and animal studies. Further records were sourced from citations within retrieved articles and via Google Scholar. For an in-depth understanding of our search strategy, including the keywords and filters, refer to the Supplemental Information ("Search Strategy").

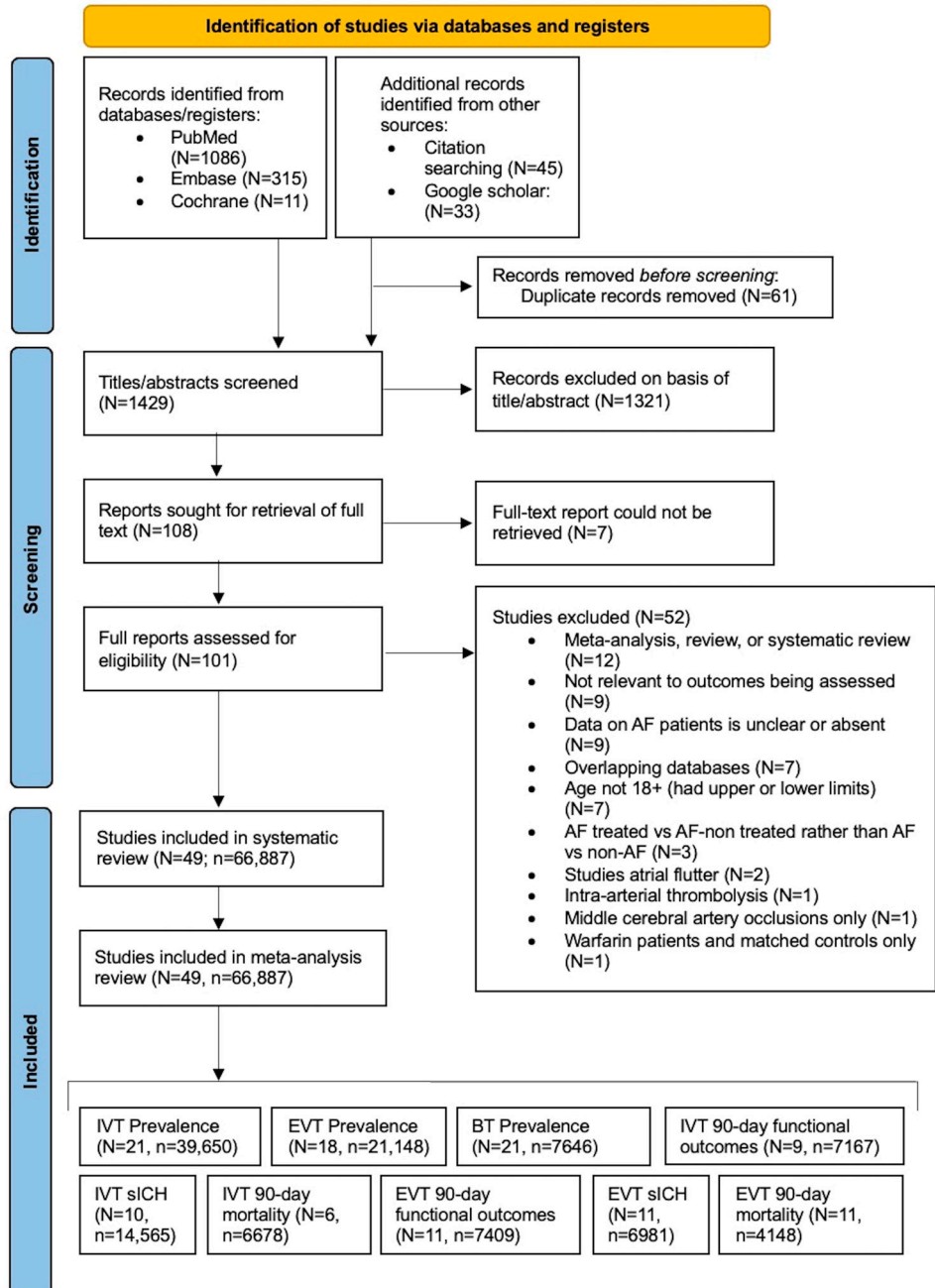

**Figure 1. PRISMA flowchart illustrating the process of study selection.** Abbreviations: N = number of studies, n = total number of patients, AF = atrial fibrillation, IVT = intravenous thrombolysis, EVT = endovascular thrombectomy, BT = bridging therapy, sICH = symptomatic intracerebral haemorrhage.

## 2.2. Inclusion and Exclusion Criteria

Studies were eligible if they met the following criteria: (1) patients diagnosed with AIS; (2) patients aged 18 years and above; (3) consecutive patients who underwent IVT and/or EVT; (4) availability of data on the prevalence of AF; and (5) studies demonstrating robust

methodological design (with a minimum sample size of 20 patients). Exclusion criteria comprised (1) animal studies; (2) studies with inaccessible full-text reports; (3) studies published in languages other than English; (4) subsequent studies from the same database; and (5) studies involving intra-arterial thrombolysis.

### 2.3. Data Extraction

The titles and abstracts of all studies from the search were initially screened using EndNote 20.5 (Clarivate, Philadelphia, PA, USA). Studies that clearly did not meet the specified eligibility criteria were excluded at this preliminary stage. Subsequently, full-text reports were obtained for the remaining studies whenever possible. These reports were thoroughly assessed to determine their eligibility for inclusion in our analysis. Meta-analyses, reviews and systematic reviews were excluded at this stage. However, reading the full texts of these studies yielded additional studies for potential inclusion in our analyses and provided valuable insights for the subsequent discussion. The screening process was conducted independently by two researchers, and any discrepancies were resolved through consensus discussions.

Data extraction sheets were prepared to capture the following details: (1) study characteristics (author, year, country, number of centres, study type); (2) reperfusion therapy type (IVT, EVT, or BT); (3) number of patients receiving reperfusion therapy with and without AF; (4) patient demographics (age, male sex, baseline National Institutes of Health Stroke Scale (NIHSS) score); (5) comorbidities (diabetes, lipid disorders, hypertension, coronary artery disease, heart failure, previous stroke/TIA, smoking); and (6) clinical outcomes (90-day functional status, sICH, 90-day mortality). Functional status was assessed using the modified Rankin Scale (mRS) [37], with mRS scores of zero-to-two indicating favourable functional outcomes and scores of three-to-six indicating poor outcomes. The 90-day timepoint was selected for functional outcomes and mortality data, as it serves as a predictive measure of long-term prognosis and is commonly available [38]. The Wan et al. [39] method was employed to approximate the mean and standard deviation (SD) from the median and interquartile range (IQR) values for age and baseline NIHSS where appropriate.

### 2.4. Assessment of Methodological Quality in Included Studies

To ensure methodological rigor, we employed the modified Jadad scale to evaluate the quality of the studies included in the meta-analysis [40]. This assessment focused on elements such as randomisation, blinding, patient withdrawals, inclusion/exclusion criteria clarity, reporting of adverse events, and statistical analysis methods [41]. The detailed evaluation of each study can be found in Supplemental Table S3. Additionally, we considered potential funding bias by examining funding source(s) and author conflicts of interest for any authors. Studies received a bias score: zero indicated low potential for bias, one or two indicated moderate potential, and three suggested high potential.

### 2.5. Statistical Analysis

Statistical analyses were conducted using STATA (Version 13.0, StataCorp, College Station, TX, USA). Meta-analyses were performed if at least four studies were available. For pooled prevalence of AF, we employed the "metaprop" package, utilising a random-effects model with refined 95% confidence intervals (95% CI) using the "cimethod (exact)" and "ftt" commands [42]. Subgroup analyses were conducted based on study type and region.

For the association between AF and clinical outcomes following reperfusion therapy, we used the STATA's "metan" package to perform meta-analyses and generate forest plots using odds ratios (ORs) as the measure of association. A random effects model was implemented with the DerSimonian–Laird (DL) method [43]. Subgroup analyses compared the prospective and retrospective studies. The "metaninf" package was utilised in STATA to examine changes in the pooled odds ratios if a single study was to be omitted. To assess for publication bias, Egger's test and funnel plots were used from the "metabias" and

"metafunnel" packages. Asymmetry in the funnel plot could indicate publication bias, further verified by the *p*-value from Egger's test.

Heterogeneity was assessed primarily using the I$^2$ statistic, with values categorised as 0–40%, 30–60%, 50–90%, and 75–100% representing low, moderate, substantial, and considerable heterogeneity, respectively [44]. Cochran's Q test *p*-values were also considered, and between-study variances were estimated using Tau-squared. A significance level of *p* < 0.05 was applied to all analyses in this study [45].

## 3. Results

The initial search across PubMed, Embase and Cochrane databases yielded a total of 1412 records (as outlined in detail in the PRISMA flowchart, Figure 1). Additionally, 78 sources were identified through Google Scholar and by handsearching the references of previous reports. After removing duplicates, the titles and abstract of 1429 studies were screened. Subsequently, 108 full-text reports were assessed, of which 101 could be successfully retrieved. Among these, 52 of these studies were excluded due to non-compliance with inclusion criteria. Finally, a rigorous selection process led to the inclusion of 49 studies for the systematic review and subsequent meta-analysis.

### 3.1. Description of Included Studies

This meta-analysis included 49 studies, comprising a total of 66,887 patients. Twenty-one studies reported on the prevalence of AF in patients receiving IVT [22–24,46–63], of which thirteen additionally reported on clinical outcomes in patients with AF compared to those without AF [22–24,46,47,49,50,53,55,58,59,61,62]. Eighteen studies [25–27,33,64–77] reported on the prevalence of AF in patients receiving EVT, of which eleven additionally reported on clinical outcomes in patients with AF compared to those without AF [25–27,64,68,69,71–73,77,78]. Nine of the EVT studies [25–27,33,68,71–73,77] also reported on the prevalence of AF in patients receiving BT. An additional nine studies reported on the prevalence of AF in patients receiving BT [28–30,79–84]. Three studies further reported on clinical outcomes in AF patients compared to non-AF patients receiving BT [28,73,81].

Table 1 presents the clinical characteristics of the 49 studies, encompassing both the total number of patients with AF and the overall total number of patients. This table provides a comprehensive overview of the diversity observed across the studies included in our meta-analysis. Notably, it highlights variations in patient demographics, study design, and geographical distribution. Further description of comorbidities such as hypertension, diabetes, and dyslipidaemia can be found in Table 2. Moreover, Table 3 presents tabulated outcomes of the clinical outcomes of patients, while Table 4 offers a concise summary of the outputs of the meta-analysis.

**Table 1.** Baseline characteristics of studies included within the meta-analysis.

| Study ID | Author | Year | Country | Centres | Study Type | Reperfusion Type | AF (n) | Overall (n) | Age ± SD [a] | | | Male (%) [a] | | | Baseline NIHSS Score ± SD [a] | | |
|---|---|---|---|---|---|---|---|---|---|---|---|---|---|---|---|---|---|
| | | | | | | | | | AF | Non-AF | Overall | AF | Non-AF | Overall | AF | Non-AF | Overall |
| 1 | Akbik et al. [64] | 2021 | International | 15 | Retrospective | EVT | 1517 | 4169 | 76 ± 11 | 65 ± 15 | - | 42.2 | 50.1 | - | 16 ± 6 | 15 ± 7 | - |
| 2 | Akbik et al. [28] | 2022 | International | 22 | Retrospective | BT | 1036 | 3140 | 76 ± 11 | 65 ± 15 | - | 46.0 | 52.0 | - | 16 ± 6 | 15 ± 7 | - |
| 3 | Al-Khaled et al. [46] | 2014 | Germany | 15 | Prospective | IVT | 387 | 1007 | - | - | 71.5 ± 12.2 | - | - | 49.6 | - | - | 11.6 ± 5.6 |
| 4 | Alobaida et al. [33] | 2023 | International | 27 | Retrospective | EVT | 1718 | 3106 | 73.6 ± 12.6 | 61.1 ± 14.8 | - | 47.1 | 57.1 | - | - | - | - |
| 4a | Alobaida et al. [33] | 2023 | International | 27 | Retrospective | BT | 549 | 889 | - | - | - | - | - | - | - | - | - |
| 5 | Awadh et al. [47] | 2010 | Scotland | 1 | Retrospective | IVT | 74 | 228 | 76 ± 10 | 66.4 ± 13.4 | - | 40.5 | 59.1 | - | 13.7 ± 8.3 | 13.8 ± 9.7 | - |
| 6 | Bavarsad Shahripour et al. [79] | 2022 | USA | 5 | Prospective | BT | 17 | 45 | 76.4 ± 11.3 | 65.4 ± 16.5 | - | 58.8 | 71.4 | - | - | - | - |
| 7 | Berkhemer et al. [65] | 2015 | The Netherlands | 16 | Prospective RCT | EVT | 66 | 233 | - | - | 65.4 ± 16.0 | - | - | 57.9 | 17.3 ± 5.2 | - | 17.3 ± 5.2 |
| 8 | Campbell et al. [80] | 2015 | Australia and New Zealand | 14 | Prospective RCT | BT | 12 | 35 | - | - | 68.6 ± 12.3 | - | - | 48.6 | 16.7 ± 5.4 | - | 16.7 ± 5.4 |
| 9 | Casetta et al. [66] | 2022 | Italy | - | Prospective | EVT | 1193 | 3422 | - | - | 70.6 | - | - | 47.4 | - | - | - |
| 10 | Chalos et al. [81] | 2019 | The Netherlands | - | Prospective | BT | 186 | 1144 | 76.3 ± 12.0 | 68 ± 15.6 | - | 46.8 | 54.8 | - | 16.3 ± 6.7 | 15 ± 5.9 | - |
| 11 | Churojana et al. [26] | 2018 | Thailand | 1 | Retrospective | EVT | 50 | 134 | 69.2 ± 12.9 | 60.2 ± 16 | - | 54.0 | 60.7 | - | 17.4 ± 5.5 | 17.1 ± 6.3 | - |
| 11a | Churojana et al. [26] | 2018 | Thailand | 1 | Retrospective | BT | 10 | 29 | - | - | - | - | - | - | - | - | - |
| 12 | Das et al. [48] | 2020 | India | 1 | Prospective | IVT | 6 | 60 | - | - | 63.9 | - | - | 56.7 | - | - | - |
| 13 | Dharmasaroja et al. [49] | 2012 | Thailand | 1 | Prospective | IVT | 46 | 194 | - | - | 64 ± 13 | - | - | 59.8 | - | - | 17.3 ± 27.6 |
| 14 | Fischer et al. [30] | 2022 | Europe and Canada | 48 | Prospective RCT | BT | 22 | 198 | - | - | 72.7 ± 11.9 | - | - | 49.8 | - | - | 16.3 ± 6.0 |
| 15 | Frank et al. [50] | 2012 | International | - | Retrospective | IVT | 639 | 3027 | 74.2 ± 9.5 | 65.7 ± 12.5 | | 47.3 | 58.2 | - | - | - | - |
| 16 | Fu et al. [27] | 2021 | Australia | 1 | Prospective | EVT | 171 | 349 | 77 ± 9.7 | 61 ± 31.4 | | 48.5 | 58.4 | - | 17.7 ± 8.2 | 16.3 ± 9.0 | - |
| 16a | Fu et al. [27] | 2021 | Australia | 1 | Prospective | BT | 53 | 132 | - | - | - | - | - | - | - | - | - |
| 17 | Goyal et al. [67] | 2015 | International | 22 | Prospective RCT | EVT | 61 | 165 | - | - | 70.7 ± 15.7 | - | - | - | - | - | 16.3 ± 5.2 |
| 18 | Huang et al. [68] | 2021 | China | - | Prospective | EVT | 123 | 245 | 73.3 ± 9.0 | 63 ± 12.8 | - | 43.9 | 68.9 | - | 16.3 ± 5.3 | 14.7 ± 4.5 | - |
| 18a | Huang et al. [68] | 2021 | China | - | Prospective | BT | 33 | 83 | - | - | - | - | - | - | - | - | - |
| 19 | Kimura et al. [51] | 2009 | Japan | - | Prospective | IVT | 44 | 85 | 77.2 ± 9 | 69.4 ± 12.5 | - | 61.4 | 70.7 | - | 17.3 ± 6.5 | 12.3 ± 7.5 | - |

**Table 1.** *Cont.*

| Study ID | Author | Year | Country | Centres | Study Type | Reperfusion Type | AF (n) | Overall (n) | Age ± SD [a] | | | Male (%) [a] | | | Baseline NIHSS Score ± SD [a] | | |
|---|---|---|---|---|---|---|---|---|---|---|---|---|---|---|---|---|---|
| | | | | | | | | | AF | Non-AF | Overall | AF | Non-AF | Overall | AF | Non-AF | Overall |
| 20 | Kurmann et al. [52] | 2018 | Switzerland | 4 | Prospective | IVT | 586 | 1775 | - | - | 69.8 | - | - | 59.2 | - | - | 11.3 |
| 21 | Lasek-Bal et al. [69] | 2022 | Poland | 1 | Retrospective | EVT | 108 | 417 | 74.9 ± 9.2 | 66.8 ± 14.5 | - | 45.4 | 54.7 | - | 14.0 ± 5.4 | 12.2 ± 6.5 | - |
| 22 | LeCouffe et al. [82] | 2021 | Netherlands, Belgium and France | 20 | Prospective RCT | BT | 63 | 266 | - | - | 69 ± 11.9 | - | - | 54.1 | - | - | 15.3 ± 7.5 |
| 23 | Lee et al. [70] | 2018 | Korea | 3 | Retrospective | EVT | 354 | 720 | - | - | 67.5 | - | - | 55.1 | - | - | 16.3 ± 6.7 |
| 24 | Leker et al. [71] | 2020 | Israel | - | Retrospective | EVT | 109 | 230 | - | - | 69.3 ± 14.7 | 39.4 | 56.2 | - | - | - | 17.1 ± 6.6 |
| 24a | Leker et al. [71] | 2020 | Israel | - | Retrospective | BT | 24 | 56 | - | - | - | - | - | - | - | - | - |
| 25 | Lin et al. [72] | 2020 | Taiwan | 1 | Retrospective | EVT | 43 | 80 | 72.6 ± 9.5 | 70.9 ± 17.3 | - | 46.5 | 57.5 | - | 17.2 ± 5.1 | 17.9 ± 6.2 | - |
| 25a | Lin et al. [72] | 2020 | Taiwan | 1 | Retrospective | BT | 24 | 39 | - | - | - | - | - | - | - | - | - |
| 26 | Lin et al. [53] | 2022 | Taiwan | 30 | Prospective | IVT | 980 | 2351 | 71.7 ± 11.9 | 66.4 ± 13 | - | 58.2 | 66.7 | - | - | - | - |
| 27a | Loo et al. [73] | 2023 | Singapore, Germany, Italy, UK, China, Taiwan | 8 | Retrospective | BT | 182 | 451 | 73.2 ± 10.3 | 65.6 ± 14.1 | - | 40.7 | 63.2 | - | 18.3 ± 8.1 | 15.9 ± 7.6 | - |
| 27 | Loo et al. [73] | 2023 | Singapore, Germany, Italy, UK, China, Taiwan | 8 | Retrospective | EVT | 314 | 705 | 73.4 ± 10.5 | 65.3 ± 14.7 | - | 43.6 | 61.9 | - | 18.4 ± 8.3 | 16.5 ± 8 | - |
| 28 | Marko et al. [54] | 2020 | Austria | 38 | Retrospective | IVT | 5452 | 18,953 | - | - | 74.9 ± 13.5 | - | - | 52.8 | - | - | 9.3 ± 7.4 |
| 29 | Mehrpour et al. [24] | 2019 | Iran | 1 | Retrospective | IVT | 24 | 118 | - | - | 66.1 ± 13.4 | - | - | 66.1 | - | - | 11.1 ± 5.1 |
| 30 | Mitchell et al. [83] | 2022 | Australia, China, New Zealand, Vietnam | 25 | Prospective RCT | BT | 34 | 147 | - | - | 69.3 ± 14.2 | - | - | 59.9 | - | - | 15 ± 7.5 |
| 31 | Mujanovic et al. [74] | 2022 | Europe and Canada | 8 | Retrospective | EVT | 1347 | 2941 | 77 ± 11.1 | 69 ± 15.6 | - | 43.5 | 54.5 | - | 15.7 ± 6.7 | 14.3 ± 7.4 | - |
| 32 | Nogueira et al. [75] | 2015 | USA | 13 | Retrospective | EVT | 413 | 1122 | - | - | 67 ± 15 | - | - | 51.9 | - | - | 16.7 ± 5.2 |
| 33 | Padjen et al. [55] | 2013 | France and Serbia | - | Prospective | IVT | 155 | 734 | 75.3 ± 12.0 | 64 ± 17.8 | - | 41.9 | 55.6 | - | 13.3 ± 7.5 | 10.7 ± 7.4 | - |
| 34 | Sanak et al. [22] | 2010 | Czech Republic | 1 | Retrospective | IVT | 66 | 157 | 68.1 ± 8.2 | 66.5 ± 13.6 | - | 57.6 | 65.9 | - | 13.3 ± 5.4 | 11 ± 5.1 | - |

**Table 1.** *Cont.*

| Study ID | Author | Year | Country | Centres | Study Type | Reperfusion Type | AF (n) | Overall (n) | Age ± SD [a] AF | Age ± SD [a] Non-AF | Age ± SD [a] Overall | Male (%) [a] AF | Male (%) [a] Non-AF | Male (%) [a] Overall | Baseline NIHSS Score ± SD [a] AF | Baseline NIHSS Score ± SD [a] Non-AF | Baseline NIHSS Score ± SD [a] Overall |
|---|---|---|---|---|---|---|---|---|---|---|---|---|---|---|---|---|---|
| 35 | Sandercock et al. [56] | 2012 | International | 156 | Prospective RCT | IVT | 473 | 1515 | - | - | - | - | - | 48.4 | - | - | - |
| 36 | Seet et al. [23] | 2011 | USA | 1 | Retrospective | IVT | 76 | 214 | 78.9 ± 9.9 | 71.5 ± 14.8 | - | 42.1 | 53.6 | - | 13 ± 4.5 | 12 ± 6.0 | - |
| 37 | Shon et al. [57] | 2016 | Korea | 4 | Prospective | IVT | 123 | 318 | - | - | - | - | - | 60.4 | - | - | 12.7 ± 6.7 |
| 38 | Smaal et al. [78] [b] | 2020 | International | - | Prospective RCT | EVT | 224 | 667 | 72.8 ± 10.1 | 63.1 ± 13.7 | - | 52.2 | 51.0 | - | 17.5 ± 4.8 | 16.5 ± 5.2 | - |
| 39 | Sung et al. [58] | 2013 | Taiwan | - | Retrospective | IVT | 72 | 143 | - | - | 68.3/64.6 | 58.3 | 64.8 | - | - | - | - |
| 40 | Tong et al. [59] | 2014 | USA | - | Retrospective | IVT | 1489 | 7193 | - | - | - | - | - | 49.5 | - | - | - |
| 41 | Tong et al. [25] | 2021 | China | 111 | Prospective | EVT | 550 | 1705 | 71 ± 10.4 | 62.3 ± 11.9 | - | 44.7 | 75.5 | - | 18 ± 5.9 | 15.7 ± 7.4 | - |
| 41a | Tong et al. [25] | 2021 | China | 111 | Prospective | BT | 145 | 513 | - | - | - | - | - | - | - | - | - |
| 42 | Vorasoot et al. [60] | 2020 | Thailand | 7 | Retrospective | IVT | 177 | 772 | - | - | 63 ± 13.2 | - | - | 54.4 | - | - | 2.5 ± 5.6 |
| 43 | Wu et al. [61] | 2022 | China | 8 | Retrospective | IVT | 242 | 630 | - | - | 72.4/61.8 | 50.0 | 69.6 | - | - | - | - |
| 44 | Yang et al. [84] | 2020 | China | 41 | Prospective RCT | BT | 149 | 329 | - | - | 68.7 ± 11.2 | - | - | 55.0 | - | - | 17.7 ± 6.0 |
| 45 | Yoshimura et al. [76] | 2018 | Japan | 46 | Prospective | EVT | 658 | 1278 | - | - | 74.7 ± 11.4 | - | - | 59.2 | - | - | 18 ± 7.4 |
| 46 | Zdraljevic et al. [77] | 2022 | Serbia | 1 | Prospective | EVT | 62 | 127 | 73.3 ± 9.5 | 59.8 ± 13.3 | - | 43.5 | 63.1 | - | 16.7 ± 5.8 | 15.5 ± 6.2 | - |
| 46a | Zdraljevic et al. [77] | 2022 | Serbia | 1 | Prospective | BT | 11 | 32 | - | - | - | - | - | 53.5 | - | - | - |
| 47 | Zhang et al. [62] | 2010 | China | - | Retrospective | IVT | 22 | 53 | 68.3 ± 8.8 | 60.7 ± 12.3 | - | 40.9 | 74.2 | - | 12 ± 7.1 | 9.1 ± 7.3 | - |
| 48 | Zhao et al. [63] | 2017 | China | - | Retrospective | IVT | 30 | 123 | - | - | 65.6 ± 11.8 | - | - | 62.6 | - | - | 7.7 ± 6 |
| 49 | Zi et al. [29] | 2021 | China | 33 | Prospective RCT | BT | 62 | 118 | - | - | 69.3 ± 13.5 | - | - | 55.9 | - | - | 16.3 ± 5.3 |

Abbreviations: AF = atrial fibrillation, n = number of patients, SD = standard deviation, NIHSS = National Institutes of Health Stroke Scale, IVT = intravenous thrombolysis, EVT = endovascular thrombectomy, BT = bridging therapy, USA = United States of America, UK = United Kingdom, RCT = randomised controlled trial. [a] Where studies reported the statistics for the AF and non-AF group separately, only these were included in the table. Where the mean age or NIHSS score was reported for two subgroups (e.g., mean age of AF patients above and below age 80), a weighted average was manually calculated, [b] This study is a pooled analysis of 6 RCTs and was included since it reports on the clinical outcomes of AF patients. However, it was not included in the meta-analyses for prevalence to avoid overlaps since some of its constituent RCTs were already present in the prevalence analyses.

**Table 2.** Rates of comorbidities within cohorts included in the meta-analysis.

| Study ID | Author | Year | Diabetes, n (%) | Lipid Disorders, n (%) | Hypertension, n (%) | CAD, n (%) | Heart Failure, n (%) [d] | Previous Stroke/TIA, n (%) | Smoking, n (%) |
|---|---|---|---|---|---|---|---|---|---|
| 1 | Akbik et al. [64] | 2021 | 1174 (28.21) | 1647 (39.53) [a] | 3112 (74.65) | - | - | 383 (12.55) [d] | - |
| 2 | Akbik et al. [28] | 2022 | 784 (25.02) | 1255 (39.99) [a] | 2247 (71.56) | - | - | 320 (12.97) [d] | - |
| 3 | Al-Khaled et al. [46] | 2014 | 198 (19.66) | 477 (47.37) [b] | 795 (78.95) | - | - | 197 (19.56) [d] | - |
| 4 | Alobaida et al. [33] | 2023 | 945 (30.42) | 1550 (51.39) [a] | 2398 (77.21) | 870 (28.01) | 906 (29.17) | 304 (9.79) [e] | - |
| 4a | Alobaida et al. [33] | 2023 | - | - | - | - | - | - | - |
| 5 | Awadh et al. [47] | 2010 | 24 (10.53) | 48 (21.05) [a] | 151 (66.23) | - | - | 45 (19.74) [f] | 56 (24.56) [i] |
| 6 | Bavarsad Shahripour et al. [79] | 2022 | 5 (11.11) | - | 28 (62.22) | 8 (17.78) | 6 (13.33) | - | 4 (8.89) [g] |
| 7 | Berkhemer et al. [65] | 2015 | 34 (14.59) | 58 (24.89) [a] | 98 (42.06) | - | - | 29 (12.45) [d] | 65 (28.89) [g] |
| 8 | Campbell et al. [80] | 2015 | 2 (5.71) | - | 21 (60.00) | - | - | - | 12 (34.29) [i] |
| 9 | Casetta et al. [66] | 2022 | 519 (15.46) | 786 (23.41) [c] | 2099 (62.51) | - | - | - | 572 (17.04) [h] |
| 10 | Chalos et al. [81] | 2019 | 197 (17.06) | - | 562 (49.08) | - | - | 164 (14.21) [d] | - |
| 11 | Churojana et al. [26] | 2018 | - | - | - | - | - | - | - |
| 11a | Churojana et al. [26] | 2018 | - | - | - | - | - | - | - |
| 12 | Das et al. [48] | 2020 | 27 (45.00) | 36 (60.00) [b] | 44 (73.33) | 7 (11.67) | - | 9 (15.00) [e] | 15 (25.00) [g] |
| 13 | Dharmasaroja et al. [49] | 2012 | 50 (25.77) | 61 (31.44) [a] | 116 (59.79) | 28 (14.43) | - | 28 (14.43) [d] | - |
| 14 | Fischer et al. [30] | 2022 | - | 71 (36.60) [b] | 118 (58.42) | - | - | 20 (9.95) [d], 14 (7.00) [e] | - |
| 15 | Frank et al. [50] | 2012 | 543 (17.94) | - | 1930 (63.76) | - | - | - | - |
| 16 | Fu et al. [27] | 2021 | 88 (25.21) | 205 (58.74) [a] | 250 (71.63) | - | 36 (10.32) | 73 (20.92) [f] | 68 (19.48) [g] |
| 16a | Fu et al. [27] | 2021 | - | - | - | - | - | - | - |
| 17 | Goyal et al. [67] | 2015 | 33 (20.00) | 58 (35.15) [a] | 105 (63.64) | 40 (24.24) | 24 (14.55) | 17 (10.30) [d] | 80 (48.48) [h] |
| 18 | Huang et al. [68] | 2021 | 37 (15.10) | 9 (3.67) [a] | 140 (57.14) | 45 (18.37) | - | 52 (21.22) [d] | 71 (28.98) [i] |
| 18a | Huang et al. [68] | 2021 | - | - | - | - | - | - | - |

**Table 2.** *Cont.*

| Study ID | Author | Year | Diabetes, n (%) | Lipid Disorders, n (%) | Hypertension, n (%) | CAD, n (%) | Heart Failure, n (%) [d] | Previous Stroke/TIA, n (%) | Smoking, n (%) |
|---|---|---|---|---|---|---|---|---|---|
| 19 | Kimura et al. [51] | 2009 | 17 (20.00) | 19 (22.35) [a] | 49 (57.65) | - | - | - | - |
| 20 | Kurmann et al. [52] | 2018 | 309 (16.66) | 910 (50.58) [a] | 1272 (68.39) | 370 (20.01) | - | - | 369 (19.79) [g] |
| 21 | Lasek-Bal et al. [69] | 2022 | 105 (25.18) | 168 (40.29) [c] | 315 (75.54) | 221 (54.17) | - | - | 115 (37.10) [i] |
| 22 | LeCouffe et al. [82] | 2021 | 50 (18.80) | 73 (27.44) [b] | 139 (52.45) | - | 15 (5.64) | 44 (16.54) [d] | 66 (25.38) [h] |
| 23 | Lee et al. [70] | 2018 | - | - | - | - | - | - | - |
| 24 | Leker et al. [71] | 2020 | 73 (31.74) | 111 (48.26) [a] | 157 (68.26) | - | - | 36 (15.65) [d] | 58 (25.22) [i] |
| 24a | Leker et al. [71] | 2020 | - | - | - | - | - | - | - |
| 25 | Lin et al. [72] | 2020 | 19 (22.89) | 34 (40.96) [a] | 53 (63.86) | 17 (20.48) | - | 18 (21.69) [d] | 21 (25.30) [i] |
| 25a | Lin et al. [72] | 2020 | - | - | - | - | - | - | - |
| 26 | Lin et al. [53] | 2022 | 755 (32.20) | 829 (35.26) [a] | 1679 (71.60) | 323 (13.77) | - | - | - |
| 27a | Loo et al. [73] | 2023 | 131 (29.05) | 188 (41.69) [c] | 326 (72.28) | 75 (16.63) | - | 57 (12.64) [f] | 52 (11.53) [i] |
| 27 | Loo et al. [73] | 2023 | 206 (29.22) | 283 (40.14) [c] | 509 (72.20) | 125 (17.73) | - | 115 (16.31) [f] | 81 (11.49) [g] |
| 28 | Marko et al. [54] | 2020 | 3957 (20.88) | 10,055 (53.05) [b] | 14,885 (78.54) | - | - | 3265 (17.23) [d] | 3099 (16.35) [g] |
| 29 | Mehrpour et al. [24] | 2019 | 41 (34.75) | 28 (23.73) [c] | 82 (69.49) | 55 (46.61) | - | 25 (21.19) [d] | 25 (21.19) [i] |
| 30 | Mitchell et al. [83] | 2022 | - | - | 89 (60.54) | - | - | 18 (12.24) [f] | - |
| 31 | Mujanovic et al. [74] | 2022 | 566 (19.54) | 1416 (49.13) [c] | 1997 (68.89) | - | - | 344 (13.89) [d] | 702 (24.98) [i] |
| 32 | Nogueira et al. [75] | 2015 | 265 (23.62) | - | 773 (68.80) | - | - | - | - |
| 33 | Padjen et al. [55] | 2013 | 122 (16.62) | 336 (45.78) [b] | 483 (65.80) | - | - | 78 (10.63) [d] | 203 (27.66) [g] |
| 34 | Sanak et al. [22] | 2010 | - | - | - | - | - | - | - |
| 35 | Sandercock et al. [56] | 2012 | - | - | - | - | - | - | - |
| 36 | Seet et al. [23] | 2011 | 28 (13.08) | 109 (50.93) [a] | 164 (76.64) | 79 (36.92) | - | 47 (21.96) [f] | 29 (13.55) [g] |
| 37 | Shon et al. [57] | 2016 | 94 (29.56) | 106 (33.33) [c] | 230 (72.33) | - | - | 54 (16.98) [d] | 77 (24.21) [g] |

**Table 2.** *Cont.*

| Study ID | Author | Year | Diabetes, n (%) | Lipid Disorders, n (%) | Hypertension, n (%) | CAD, n (%) | Heart Failure, n (%) [d] | Previous Stroke/TIA, n (%) | Smoking, n (%) |
|---|---|---|---|---|---|---|---|---|---|
| 38 | Smaal et al. [78] | 2020 | 114 (17.09) | - | 369 (55.32) | - | - | - | - |
| 39 | Sung et al. [58] | 2013 | 48 (33.57) | 83 (58.04) [a] | 112 (78.32) | - | - | 27 (18.88) [d] | 34 (23.78) [g] |
| 40 | Tong et al. [59] | 2014 | 1793 (24.93) | 3020 (41.99) [c] | 5757 (80.04) | - | 751 (10.44) | 1251 (17.39) [d] | 1537 (21.37) [i] |
| 41 | Tong et al. [25] | 2021 | 324 (19.00) | - | 1006 (59.00) | - | - | 337 (19.77) [d] | 706 (41.41) [h] |
| 41a | Tong et al. [25] | 2021 | - | - | - | - | - | - | - |
| 42 | Vorasoot et al. [60] | 2020 | 163 (21.11) | 137 (17.75) [c] | 378 (48.96) | 52 (6.74) | 11 (1.42) | 118 (15.28) [d], 18 (2.33) [e] | 281 (36.40) [i] |
| 43 | Wu et al. [61] | 2022 | 106 (16.83) | 216 (34.29) [a] | 481 (76.35) | 34 (5.40) | - | 98 (15.56) [d] | 221 (35.08) [g] |
| 44 | Yang et al. [84] | 2020 | 65 (19.76) | 14 (4.26) [b] | 201 (61.09) | - | 17 (5.17) | 47 (14.29) [d] | 68 (20.67) [i] |
| 45 | Yoshimura et al. [76] | 2018 | 236 (18.47) | - | 739 (57.82) | - | - | 95 (7.43) [d] | 183 (14.32) [g] |
| 46 | Zdraljevic et al. [77] | 2022 | 22 (17.32) | 75 (59.06) [a] | 102 (80.31) | - | - | 19 (14.96) [d] | 32 (25.20) [i] |
| 46a | Zdraljevic et al. [77] | 2022 | - | - | - | - | - | - | - |
| 47 | Zhang et al. [62] | 2010 | 5 (9.43) | - | 26 (49.06) | - | - | - | - |
| 48 | Zhao et al. [63] | 2017 | 26 (21.14) | - | 83 (67.48) | 14 (11.38) | - | 40 (32.52) [d] | 49 (39.84) [i] |
| 49 | Zi et al. [29] | 2021 | 20 (16.95) | 22 (18.64) [a] | 74 (62.71) | 19 (16.10) | - | 19 (16.10) [d] | 29 (24.58) [h] |

Abbreviations: TIA = transient ischaemic attack, CAD = coronary artery disease, n = number of patients. [a]: described as hyperlipidaemia. [b]: described as hypercholesterolaemia. [c]: described as dyslipidaemia. [d]: prior stroke only. [e]: prior TIA only. [f]: both stroke and TIA. [g]: current smokers only. [h]: previous smokers as well as current smokers. [i]: does not specify if smokers are current or previous smokers.

**Table 3.** Clinical outcomes of studies within the meta-analysis.

| Study ID | Author | Reperfusion Therapy | Good Functional Outcome at 90 Days | | sICH | | 90-Day Mortality | | sICH Definition |
|---|---|---|---|---|---|---|---|---|---|
| | | | AF, n (%) | No AF, n (%) | AF, n (%) | No AF, n (%) | AF, n (%) | No AF, n (%) | |
| 1 | Akbik et al. [64] | EVT | 426 (31.14) | 1029 (42.28) | 89 (7.64) | 160 (7.28) | 354 (25.88) | 408 (16.76) | ECASS II |
| 2 | Akbik et al. [28] | BT | 295 (33.56) | 822 (46.13) | 91 (9.15) | 140 (7.02) | 222 (25.26) | 324 (18.18) | ECASS II |
| 3 | Al-Khaled et al. [46] | IVT | - | - | 29 (7.49) | 29 (4.68) | - | - | Any bleeding that was not detected on a previous CT scan and associated with an increase in NIHSS score of $\geq$4 |
| 5 | Awadh et al. [47] | IVT | - | - | 3 (4.05) | 7 (4.55) | - | - | Deterioration in NIHSS score of $\geq$4 within 72 h, and PH1 or PH2 present on CT |
| 10 | Chalos et al. [81] | BT | 52 (29.89) | 373 (42.78) | 11 (5.91) | 56 (5.85) | 65 (34.95) | 204 (21.29) | Heidelberg Bleeding Classification |
| 11 | Churojana et al. [26] | EVT | 19 (38.00) | 32 (38.10) | 6 (12.00) | 11 (13.10) | 10 (20.00) | 16 (19.05) | NR |
| 13 | Dharmasaroja et al. [49] | IVT | - | - | 6 (13.04) | 5 (3.42) | - | - | NINDS |
| 15 | Frank et al. [50] | IVT | 211 (33.02) | 1179 (49.37) | 17 (2.66) | 41 (1.72) | 139 (21.75) | 325 (13.61) | 24-h increase in NIHSS score by $\geq$4 or any stroke/ICH leading to death |
| 16 | Fu et al. [27] | EVT | 82 (47.95) | 85 (47.75) | 2 (1.17) | 7 (3.93) | 37 (21.64) | 32 (17.98) | SITS-MOST |
| 18 | Huang et al. [68] | EVT | 27 (38.57) | 31 (44.29) | 9 (12.86) | 5 (7.14) | 15 (21.43) | 14 (20.00) | ICH with a 24-h increase in NIHSS score of $\geq$4 |
| 21 | Lasek-Bal et al. [69] | EVT | 28 (25.93) | 89 (28.80) | 7 (6.48) | 15 (4.85) | 30 (27.78) | 72 (23.30) | ECASS II |
| 24 | Leker et al. [71] | EVT | 27 (24.77) | 51 (42.15) | 6 (5.50) | 5 (4.13) | 12 (11.01) | 23 (19.01) | ECASS III |
| 25 | Lin et al. [72] | EVT | 24 (55.81) | 7 (17.50) | 3 (6.98) | 4 (10.00) | 4 (9.30) | 6 (15.00) | SITS-MOST |
| 26 | Lin et al. [53] | IVT | 351 (39.35) | 574 (47.87) | 12 (1.22) | 14 (1.02) | 93 (10.43) | 106 (8.84) | SITS-MOST |
| 27 | Loo et al. [73] | EVT | 106 (34.30) | 146 (38.52) | 30 (9.55) | 51 (13.04) | 59 (19.09) | 63 (16.62) | SITS-MOST |
| 27a | Loo et al. [73] | BT | 63 (35.00) | 118 (45.21) | - | - | 34 (18.89) | 41 (15.71) | - |

**Table 3.** *Cont.*

| Study ID | Author | Reperfusion Therapy | Good Functional Outcome at 90 Days | | sICH | | 90-Day Mortality | | sICH Definition |
|---|---|---|---|---|---|---|---|---|---|
| | | | AF, n (%) | No AF, n (%) | AF, n (%) | No AF, n (%) | AF, n (%) | No AF, n (%) | |
| 29 | Mehrpour et al. [24] | IVT | 5 (20.83) | 55 (58.51) | - | - | - | - | - |
| 33 | Padjen et al. [55] | IVT | 74 (47.74) | 375 (64.77) | - | - | 34 (21.94) | 52 (8.98) | - |
| 34 | Sanak et al. [22] | IVT | 33 (50.00) | 66 (72.53) | 3 (4.55) | 0 (0.00) | - | - | ECASS II |
| 36 | Seet et al. [23] | IVT | 32 (42.11) | 77 (55.80) | 10 (13.16) | 7 (5.07) | - | - | Haemorrhagic transformation associated with an increase in NIHSS score of ≥4 |
| 38 | Smaal et al. [78] | EVT | 95 (42.41) | 213 (48.19) | 8 (3.57) | 20 (4.51) | 46 (20.54) | 58 (13.12) | Various |
| 39 | Sung et al. [58] | IVT | 34 (47.22) | 24 (33.80) | 6 (8.33) | 66 (9.86) | 4 (5.56) | 8 (11.27) | ECASS II |
| 40 | Tong et al. [59] | IVT | - | - | 98 (6.58) | 225 (3.94) | - | - | CT showing intracranial haemorrhage and medical records noting clinical deterioration due to haemorrhage |
| 41 | Tong et al. [25] | EVT | 160 (41.34) | 155 (40.16) | 36 (9.45) | 35 (9.07) | 63 (16.28) | 71 (18.39) | Heidelberg Bleeding Classification |
| 43 | Wu et al. [61] | IVT | 97 (40.08) | 279 (71.91) | - | - | 39 (16.12) | 22 (5.67) | - |
| 46 | Zdraljevic et al. [77] | EVT | 19 (30.65) | 35 (53.85) | 5 (8.06) | 4 (6.15) | 22 (35.48) | 11 (16.92) | SITS-MOST |
| 47 | Zhang et al. [62] | IVT | 9 (40.91) | 17 (54.84) | 4 (18.18) | 2 (6.45) | 4 (18.18) | 3 (9.68) | ECASS III |

Abbreviations: n = number of patients, EVT = endovascular thrombectomy, BT = bridging therapy, IVT = intravenous thrombolysis, AF = atrial fibrillation, sICH = symptomatic intracerebral haemorrhage, ECASS = European Cooperative Acute Stroke Study, NIHSS = National Institutes of Health Stroke Scale, CT = computed tomography, PH1 = parenchymal haematoma type 1, PH2 = parenchymal haematoma type 2, NR = not reported, NINDS = National Institute of Neurological Disorders and Stroke, ICH = intracerebral haemorrhage, SITS-MOST = Safe Implementation of Thrombolysis in Stroke-Monitoring Study.

**Table 4.** Summary effect sizes and heterogeneity from the meta-analysis.

| Outcome | Reperfusion Therapy | Effect Measure | Effect Measure (95% CI) | Test of ES = 0 | Tests of Overall Effect | Heterogeneity | | | | | Heterogeneity Variance Estimates |
|---|---|---|---|---|---|---|---|---|---|---|---|
| | | | | | | Chi-Squared | Cochran's Q | *p*-Value | H (95% CI) | I² (%) (95% CI) | tau² |
| Prevalence | IVT | Prevalence | 0.31 (0.28 to 0.35) | *p* < 0.01 z = 30.02 | - | 738.91 | - | <0.01 | - | 97.29 | 0.03 |
| | EVT | Prevalence | 0.42 (0.38 to 0.46) | *p* < 0.01 z = 30.94 | - | 606.22 | - | <0.01 | - | 97.20 | 0.03 |
| | BT | Prevalence | 0.36 (0.28 to 0.43) | *p* < 0.001 z = 14.86 | - | 630.87 | - | <0.01 | - | 97.31 | 0.10 |
| 90-day mRS 0–2 | IVT | OR | 0.512 (0.376 to 0.696) | - | *p* < 0.001 z = −4.271 | - | 43.85 | <0.001 | 2.341 (1.000 to 3.897) | 81.8 (<0.1 to 93.4) | 0.1457 |
| | EVT | OR | 0.826 (0.651 to 1.049) | - | *p* = 0.117 z = −1.568 | - | 38.44 | <0.001 | 1.961 (1.000 to 3.050) | 74.0 (<0.1 to 89.2) | 0.1009 |
| sICH | IVT | OR | 1.690 (1.400 to 2.039) | - | *p* < 0.001 z = 5.473 | - | 8.14 | 0.520 | 0.951 (1.000 to 1.387) | <0.1 (<0.1 to 48.0) | <0.0001 |
| | EVT | OR | 0.982 (0.815 to 1.184) | - | *p* = 0.851 z = −0.188 | - | 7.23 | 0.703 | 0.851 (1.000 to 1.254) | <0.1 (<0.1 to 36.4) | <0.0001 |
| 90-day mortality | IVT | OR | 1.799 (1.218 to 2.657) | - | *p* = 0.003 z = 2.953 | - | 19.34 | 0.002 | 1.966 (1.000 to 3.432) | 74.1 (<0.1 to 91.5) | 0.1407 |
| | EVT | OR | 1.236 (0.969 to 1.578) | - | *p* = 0.088 z = 1.706 | - | 26.14 | 0.004 | 1.617 (1.000 to 2.465) | 61.7 (<0.1 to 83.5) | 0.0857 |

Abbreviations: IVT = intravenous thrombolysis, EVT = endovascular thrombectomy, BT = bridging therapy, mRS = modified Rankin Scale, sICH = symptomatic intracerebral haemorhage, OR = odds ratio, ES = effect size, CI = confidence interval, H = relative excess in Cochran's Q over its degrees-of-freedom, I² = proportion of total variation in effect estimate due to between-study heterogeneity (based on Q).

### 3.2. Prevalence of AF in Patients Treated with IVT

Twenty-one studies, comprising a total of 39,650 patients, were included in the final meta-analysis for the prevalence of AF in patients receiving IVT (Figures 2 and 3) [22–24,46–63]. The meta-analysis revealed an estimated prevalence of 31% (ES 0.31 [95% CI 0.28 to 0.35], *p* < 0.01). Considerable heterogeneity was found between the studies (I² = 97.3%, *p* < 0.01). The crude prevalence of AF was 28.2%, which was obtained by dividing the total number of AF patients receiving IVT by the overall cohort of patients receiving IVT in this analysis.

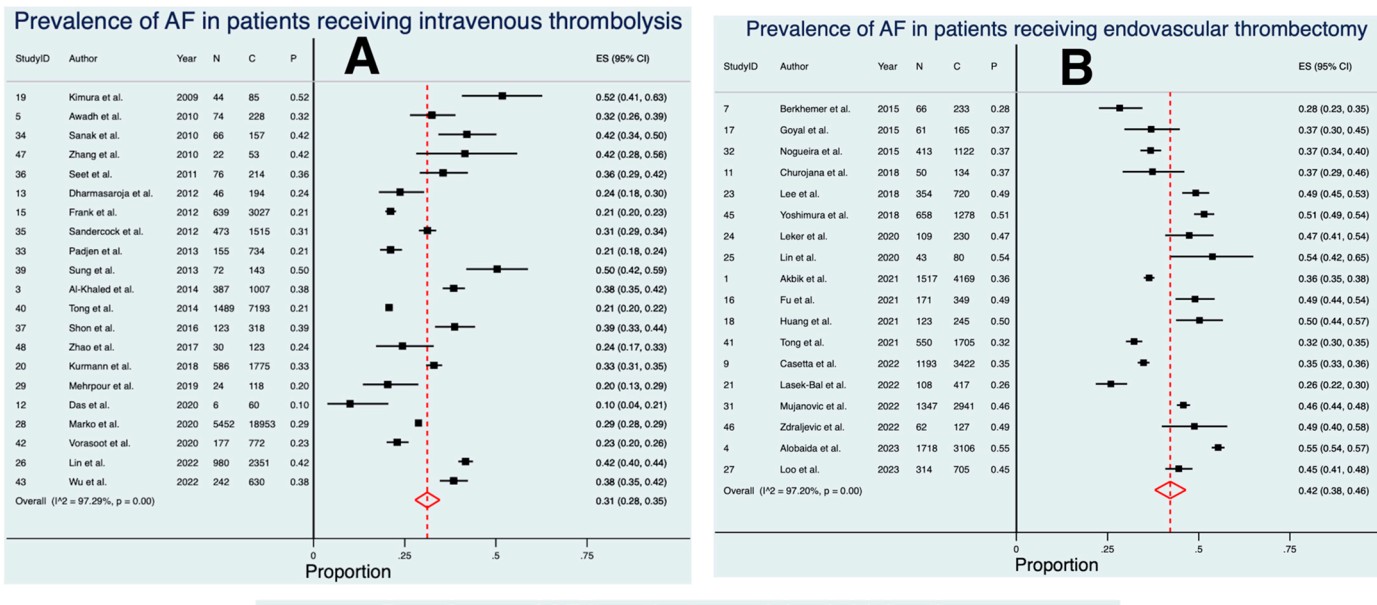

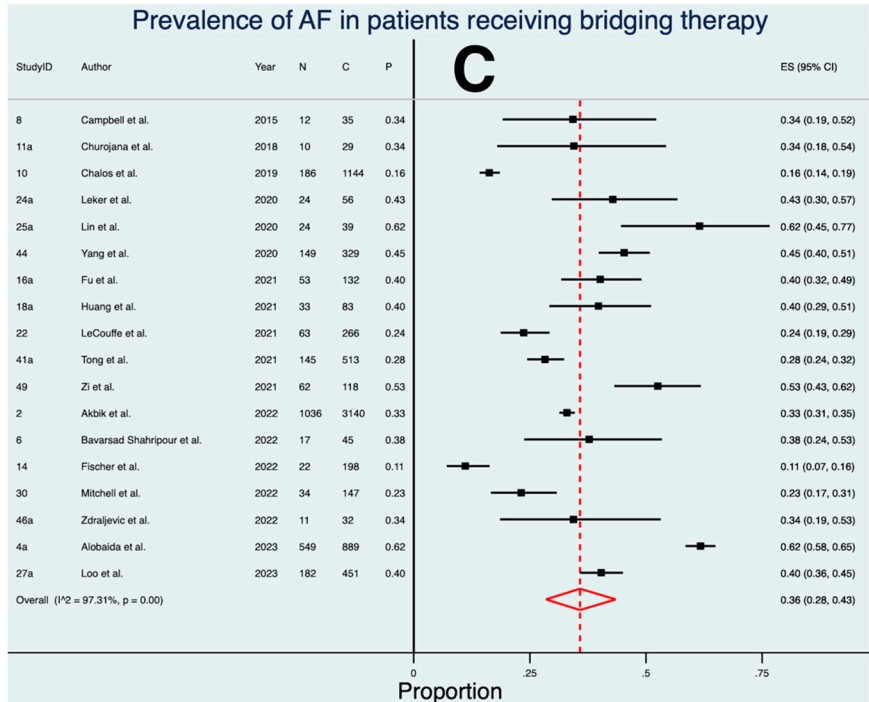

**Figure 2.** Forest plots of the pooled prevalence of atrial fibrillation in acute ischaemic stroke patients receiving reperfusion therapy. (**A**) intravenous thrombolysis, (**B**) endovascular thrombectomy, and (**C**) bridging therapy. Abbreviations: AF = atrial fibrillation, ES = effect size, CI = confidence interval, N = number of patients with AF, C = total number of patients, P = prevalence.

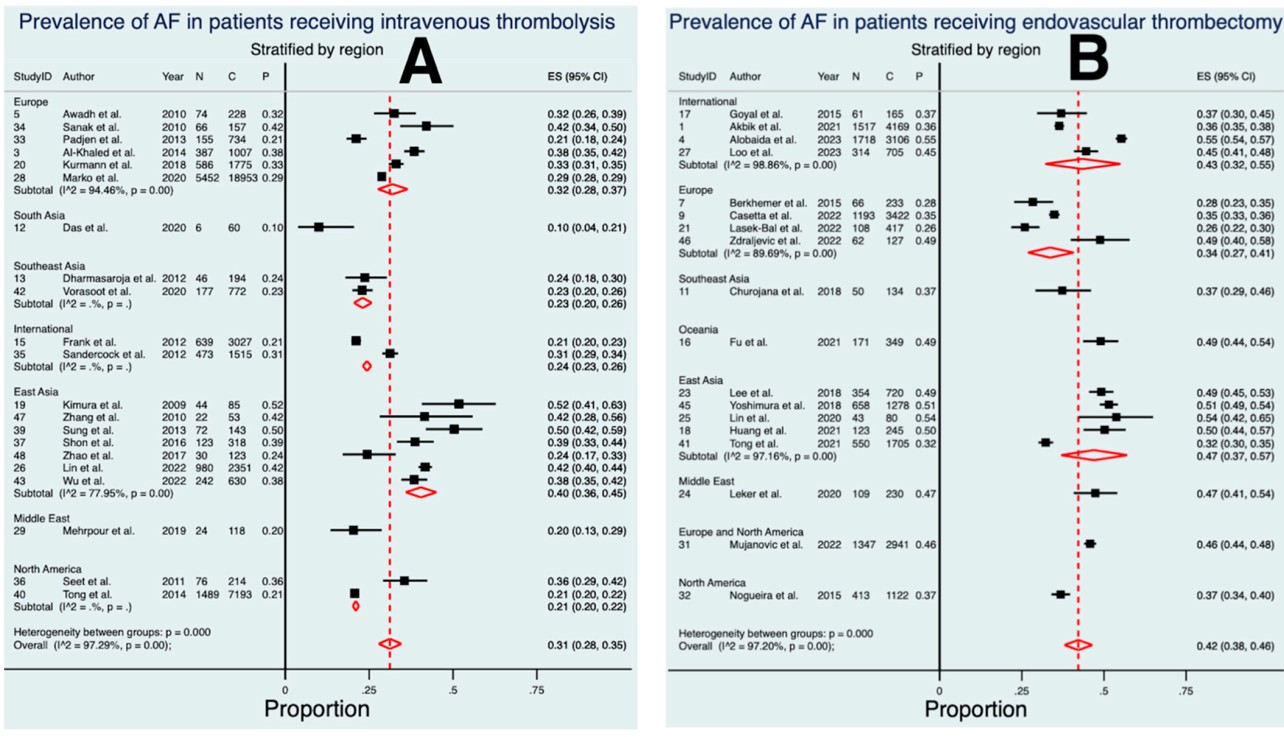

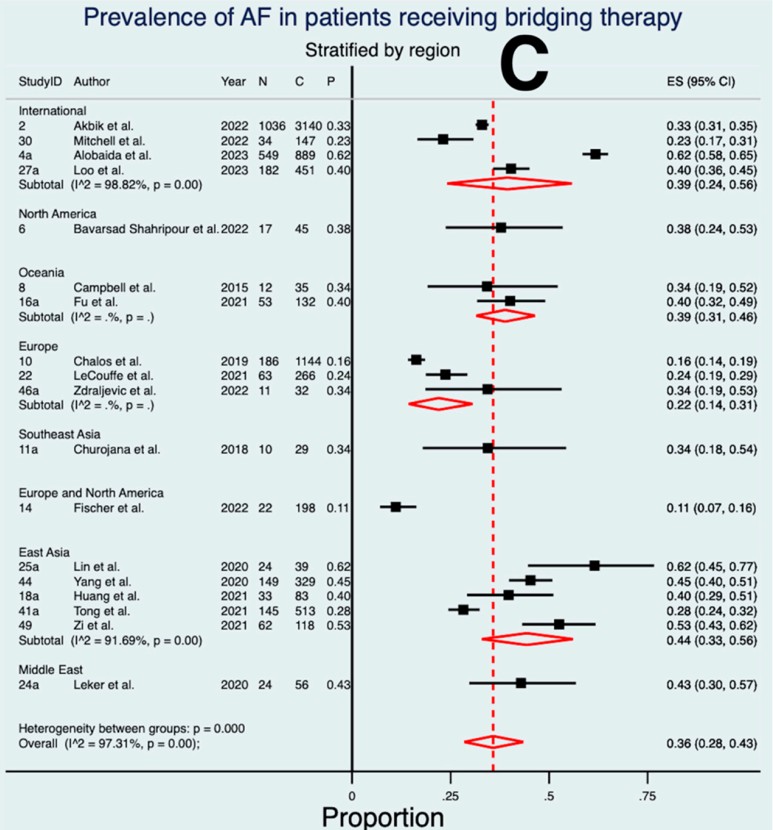

**Figure 3.** Forest plots of the estimated pooled prevalence of atrial fibrillation in acute ischaemic stroke patients receiving each type of reperfusion therapy, stratified by region. (**A**) intravenous thrombolysis, (**B**) endovascular thrombectomy, and (**C**) bridging therapy. Abbreviations: AF = atrial fibrillation, ES = effect size, CI = confidence interval, N = number of patients with AF, C = total number of patients, P = prevalence.

### 3.3. Prevalence of AF in Patients Treated with EVT

Eighteen studies, comprising a total of 21,148 patients, were included in the final meta-analysis for the prevalence of AF in patients receiving EVT (Figures 2 and 3) [25–27,33,64–77]. The meta-analysis revealed an estimated pooled prevalence of 42% (ES 0.42 [95% CI 0.38 to 0.46], $p < 0.01$). Considerable heterogeneity was found between the studies ($I^2 = 97.2\%$, $p < 0.01$). The crude prevalence of AF was 41.9%, which was obtained by dividing the total number of AF patients receiving EVT by the overall cohort of patients receiving EVT in this analysis.

### 3.4. Prevalence of AF in Patients Treated with BT

Eighteen studies, comprising a total of 7646 patients, were included in the final meta-analysis for the prevalence of AF in patients receiving BT (Figures 2 and 3) [25–30,33,68,71–73,77,79–84]. The meta-analysis revealed an estimated pooled prevalence of 36% (ES 0.36 [95% CI 0.28 to 0.43], $p < 0.01$). Considerable heterogeneity was found between the studies ($I^2 = 97.3\%$, $p < 0.01$). The crude prevalence of AF was 34.2%, which was obtained by dividing the total number of AF patients receiving BT by the overall cohort of patients receiving BT in this analysis.

### 3.5. Association between AF and Favourable 90-Day Functional Outcomes Following IVT

Nine studies, comprising a total of 7167 patients, were included in the final meta-analysis for the association between AF and favourable 90-day functional outcomes following IVT (Figure 4) [22–24,50,53,55,58,61,62]. Overall, the meta-analysis revealed that AF was associated with significantly lower odds of favourable functional outcomes at 90 days following IVT (OR 0.512 [95% CI 0.376 to 0.696], $p < 0.001$). Substantial to considerable heterogeneity was found between the studies ($I^2 = 81.8\%$, [95% CI < 0.1% to 93.4%], $p < 0.001$). Visual inspection of the funnel plot (Supplemental Figure S1) and the output from Egger's test (Supplemental Table S4) did not provide evidence of significant publication bias.

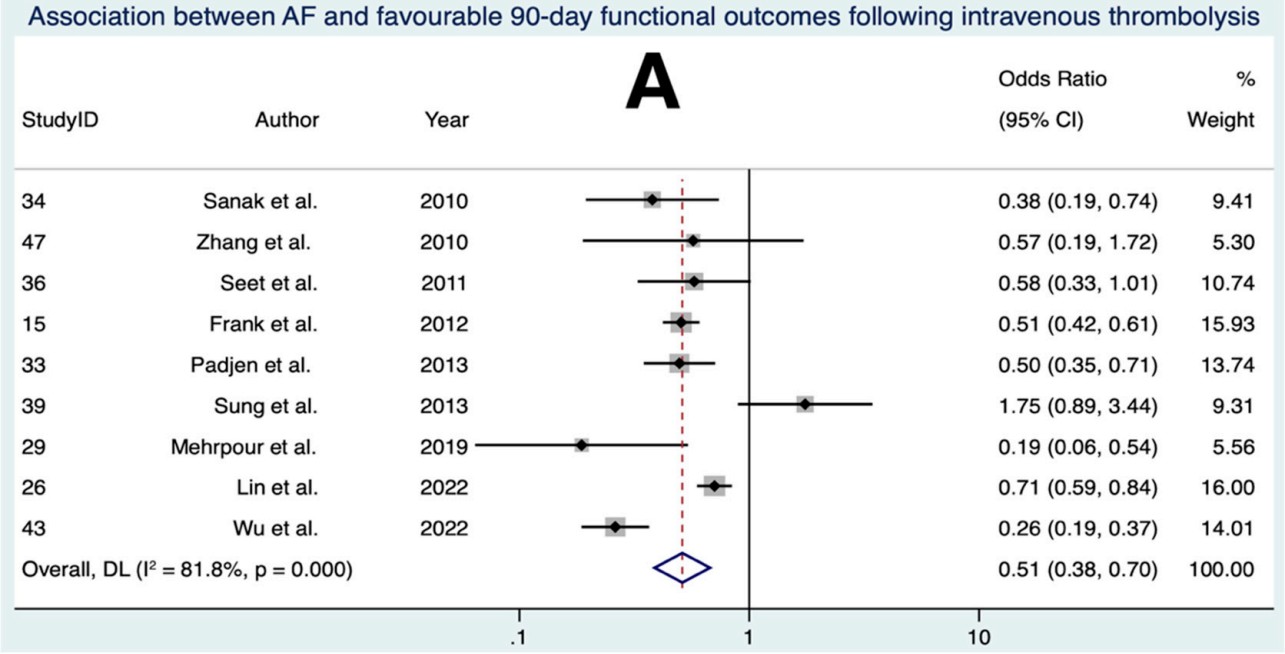

**Figure 4.** *Cont.*

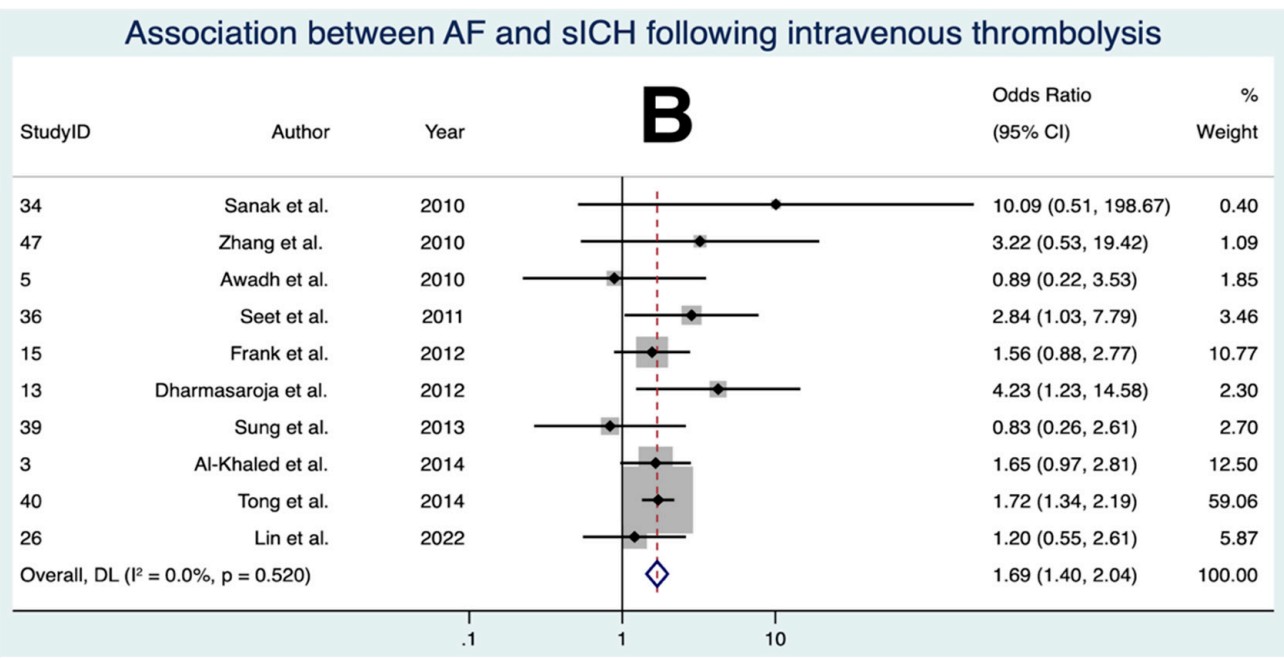

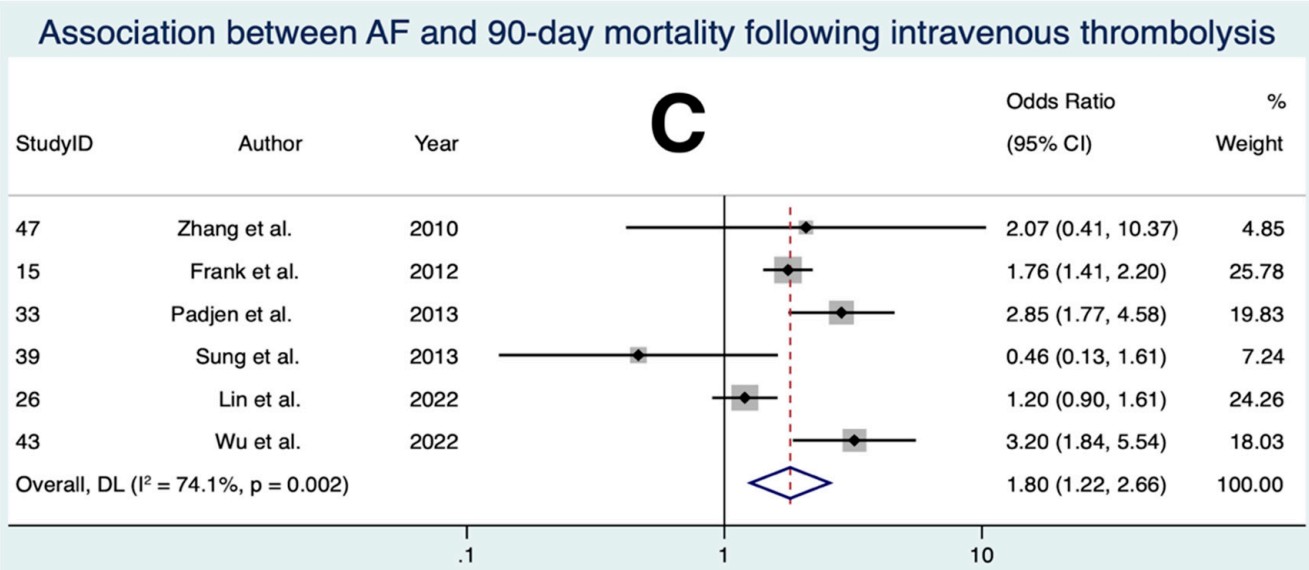

**Figure 4. Forest plots of the association between atrial fibrillation and outcomes following intravenous thrombolysis.** (**A**) association between atrial fibrillation and favourable 90-day functional outcomes. (**B**) association between atrial fibrillation and symptomatic intracerebral haemorrhage. (**C**) association between atrial fibrillation and 90-day mortality. Abbreviations: AF = atrial fibrillation, CI = confidence interval, sICH = symptomatic intracerebral haemorrhage, DL = DerSimonian–Laird.

*3.6. Association between AF and sICH Following IVT*

Ten studies, comprising a total of 14,565 patients, were included in the final meta-analysis for the association between AF and sICH following IVT (Figure 4) [22,23,46,47,49,50,53,58,59,62]. The definition of sICH used in each study is outlined in Table 3. Overall, the meta-analysis revealed that AF was associated with significantly higher odds of sICH following IVT (OR 1.690 [95% CI 1.400 to 2.039], *p* = 0.851). Low heterogeneity was found between the studies ($I^2$ = 0.0% [95% CI < 0.1% to 48.0%], *p* = 0.520). Visual inspection of the funnel plot (Supplemental Figure S1) and the output from Egger's test (Supplemental Table S4) did not provide evidence of significant publication bias.

### 3.7. Association between AF and 90-Day Mortality Following IVT

Six studies, comprising a total of 6678 patients, were included in the final meta-analysis for the association between AF and 90-day mortality following IVT (Figure 4) [50,53,55,58,61,62]. Overall, the meta-analysis revealed that AF was associated with significantly higher odds of mortality at 90 days following IVT (OR 1.799 [95% CI 1.218 to 2.657], *p* = 0.003). Substantial heterogeneity was found between the studies ($I^2$ = 74.1% [95% CI < 0.1% to 91.5%], *p* = 0.002). Visual inspection of the funnel plot (Supplemental Figure S1) and the output from Egger's test (Supplemental Table S4) did not provide evidence of significant publication bias.

### 3.8. Association between AF and Favourable 90-Day Functional Outcomes Following EVT

Eleven studies, comprising a total of 7409 patients, were included in the final meta-analysis for the association between AF and favourable 90-day functional outcomes following EVT (Figure 5) [25–27,64,68,69,71–73,77,78]. Overall, the meta-analysis revealed no significant association between AF and favourable functional outcomes at 90 days following EVT (OR 0.826 [95% CI 0.651 to 1.049], *p* = 0.117). Substantial heterogeneity was found between the studies ($I^2$ = 74.0%, [95% CI < 0.1% to 89.2%], *p* < 0.001). Visual inspection of the funnel plot (Supplemental Figure S1) and the output from Egger's test (Supplemental Table S4) did not provide evidence of significant publication bias.

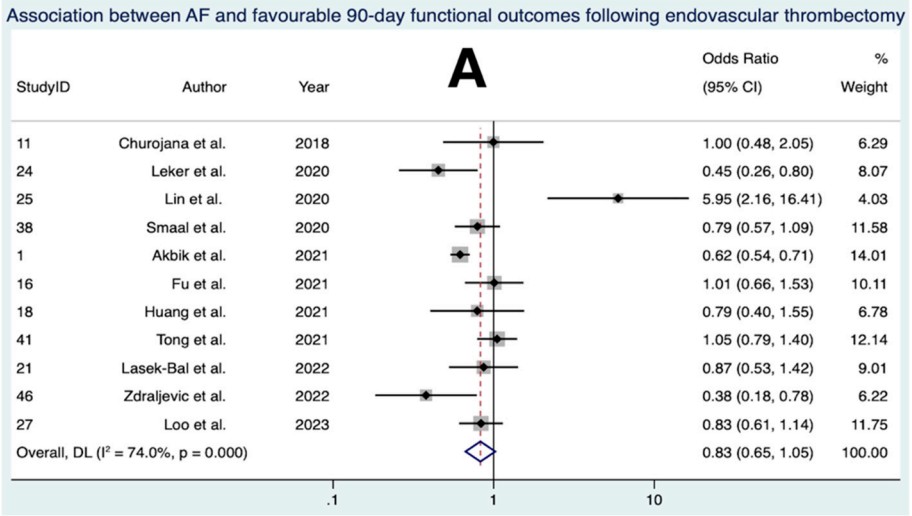

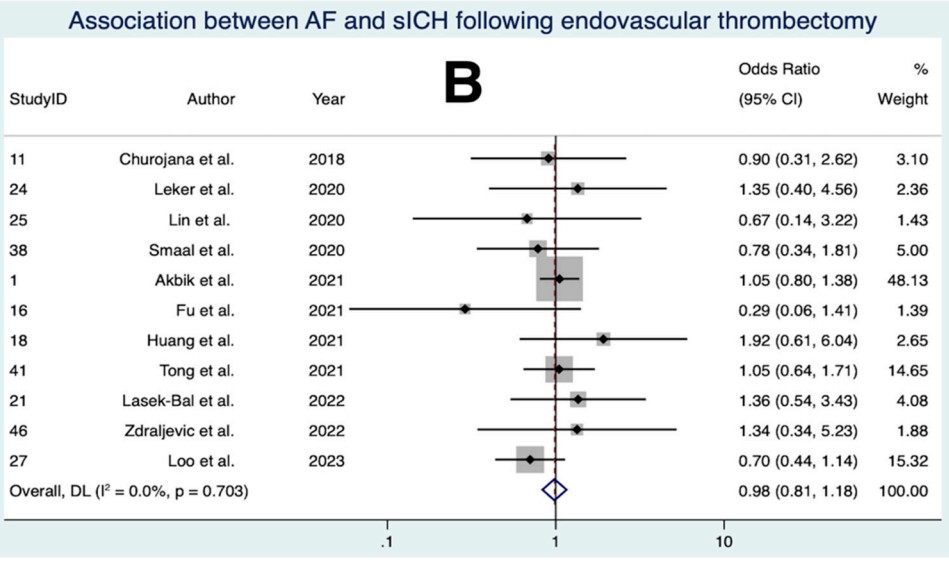

**Figure 5.** *Cont.*

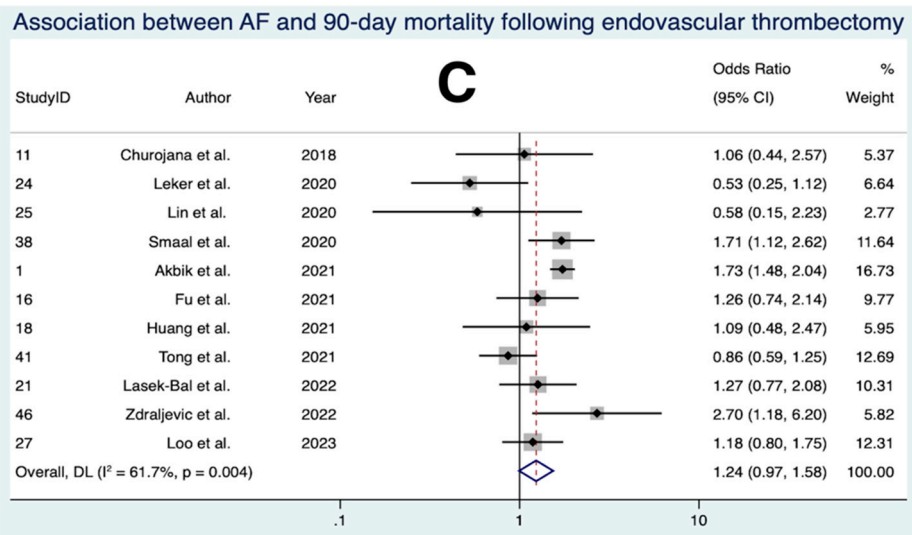

**Figure 5.** **Forest plots of the association between atrial fibrillation and outcomes following endovascular thrombectomy.** (**A**) association between atrial fibrillation and favourable 90-day functional outcomes. (**B**) association between atrial fibrillation and symptomatic intracerebral haemorrhage. (**C**) association between atrial fibrillation and 90-day mortality. Abbreviations: AF = atrial fibrillation, CI = confidence interval, sICH = symptomatic intracerebral haemorrhage, DL = DerSimonian–Laird.

### 3.9. Association between AF and sICH Following EVT

Eleven studies, comprising a total of 6981 patients, were included in the final meta-analysis for the association between AF and sICH following EVT (Figure 5) [25–27,64,68,69,71–73,77,78]. The definition of sICH used in each study is described in Table 3. Overall, the meta-analysis revealed no significant association between AF and sICH following EVT (OR 0.982 [95% CI 0.815 to 1.184], $p = 0.851$). Low heterogeneity was found between the studies ($I^2 < 0.1\%$ [95% CI < 0.1% to 36.4%], $p = 0.703$). Visual inspection of the funnel plot (Supplemental Figure S1) and the output from Egger's test (Supplemental Table S4) did not provide evidence of significant publication bias.

### 3.10. Association between AF and 90-Day Mortality Following EVT

The final meta-analysis encompassed eleven studies involving a total of 7409 patients and investigated the association between AF and 90-day mortality post EVT (Figure 5) [25–27,64,68,69,71–73,77,78]. Overall, the meta-analysis revealed no statistically significant association between AF and mortality at 90 days post EVT (OR 1.236 [95% CI 0.969 to 1.578], $p = 0.088$). Notably, substantial heterogeneity existed among the studies ($I^2 = 61.7\%$ [95% CI < 0.1% to 83.5%], $p = 0.004$). Both the funnel plot examination (Supplemental Figure S1) and the output from Egger's test (Supplemental Table S4) did not provide evidence of significant publication bias.

### 3.11. Association between AF and Favourable 90-Day Functional Outcomes Following BT

Three studies, encompassing a total of 4148 patients, investigated the association between AF and favourable 90-day functional outcomes following BT [28,73,81]. In each of these three studies, AF patients reported a decreased rate of favourable 90-day functional outcomes [28,73,81]. However, a meta-analysis could not be performed due to an insufficient number of available studies.

### 3.12. Association between AF and sICH Following BT

Two studies, comprising a total of 4133 patients, reported on the association between AF and sICH following BT [28,81]. The precise definition of sICH employed in each of these studies is outlined in Table 3. Notably, both studies observed a higher incidence of

sICH among patients with AF patients [28,81]. Nonetheless, due to scarcity of available studies, a comprehensive meta-analysis could not be undertaken.

*3.13. Association between AF and 90-Day Mortality Following BT*

Three studies, comprising a combined total of 4246 patients, reported on the association between AF and 90-day mortality following BT [28,73,81]. Across all three studies, individuals with AF consistently demonstrated a heightened 90-day mortality rate [28,73,81]. Unfortunately, a comprehensive meta-analysis could not be performed due to limited availability of studies for inclusion.

## 4. Discussion

Our study demonstrated a high prevalence of AF among patients undergoing reperfusion therapy for AIS. To the best of our knowledge, this is the first meta-analysis that estimates the pooled prevalence of AF specifically in AIS patients receiving reperfusion therapy. Whilst the pooled prevalence of AF was high for each form of reperfusion therapy, the highest prevalence was observed in patients undergoing EVT, followed by BT and IVT. Stratifying by region revealed further variations in prevalence. In the context of IVT, AF was associated with significantly lower odds of favourable functional outcomes and higher odds of sICH and mortality in AIS patients. However, no such associations between AF and these clinical outcomes were identified following EVT.

Whilst no previous meta-analyses have investigated AF prevalence in stroke patients receiving reperfusion therapy, comparisons can be drawn with studies encompassing the broader ischaemic stroke population. Interestingly, our analysis suggests a prevalence estimate towards the higher end when contrasted with nationwide studies that reported AF prevalence ranging from 18.2% to 38.0% [85–89]. However, direct comparisons with nationwide studies are potentially misleading due to global variation in AF prevalence [90,91]. Furthermore, our analysis employed a random effects model to determine pooled prevalence [42], thereby mitigating the impact of outliers and studies with disproportionately large sample sizes. Given the established association between AF and increased stroke severity [92], it is plausible that AF patients might have greater eligibility for therapies like EVT when compared to the general stroke population, considering EVT's recommendation for patients with baseline NIHSS scores of six or more [17]. Therefore, a prudent future direction is to conduct a meta-analysis on the pooled prevalence of AF in the broader ischaemic stroke population. If the prevalence of AF indeed proves higher among AIS patients receiving reperfusion therapy, this would potentially warrant increased resource allocation towards AF detection and treatment within thrombolysis and thrombectomy centres.

When stratified by region, a consistently higher prevalence of AF was observed in East Asian nations (Figure 3). However, genetic variation is unlikely to explain this alone, as studies from the USA indicate a lower AF prevalence among East Asians compared to Caucasians [93–95]. Moreover, the higher prevalence in East Asian nations such as Japan can be attributed to their ageing populations, as the prevalence of AF increases with age, aligning with the higher prevalence seen in the patients undergoing reperfusion therapy [96]. Notably, several studies from less developed nations reported a relatively lower AF prevalence, particularly in the context of IVT [24,48,49,60]. This difference could be influenced by limited access to AF diagnostic technologies in these regions, although conclusive insights cannot be drawn due to the lack of relevant studies [97]. Consequently, future epidemiological AF research should prioritise investigating these developing regions.

While no meta-analyses have directly compared AF prevalence among different forms of reperfusion therapy, the observed variation aligns with the existing literature and clinical guidelines. It is logical that AF prevalence is lowest in IVT and highest in EVT, considering pertinent factors. AF patients often receive oral anticoagulant therapy [45,98] which can influence clotting markers [99,100] and contraindicate IVT due to bleeding risks [17]. EVT is recommended for large vessel occlusion (LVO) strokes [17], which often have a cardioembolic source, frequently AF, explaining the higher AF prevalence in patients receiving

EVT [27,101]. This implies that AF suspicion is heightened in EVT-treated patients with unknown stroke aetiology, necessitating adequate cardiac monitoring [101].

Our study reaffirms that AF is associated with worse clinical outcomes following IVT, aligning with existing meta-analyses [34,102]. Two recent studies from 2022 were added to our analysis [53,61]. Some studies were excluded for reasons such as an inability to extract the precise numbers for each outcome [51,63] or the follow-up not being at 90 days [103]. In the broad context, our findings underscore the negative association between AF and clinical outcomes following IVT. It is vital that clinicians are aware of this, as it may influence their management plan and how they communicate the risk of treatments with patients. A pathophysiological hypothesis for this association is that the cardiac emboli induced by AF may exhibit greater resistance to IVT [104] due to the thrombi containing a smaller proportion of fibrin compared to erythrocytes [105]. However, the suggestion that cardiac emboli have a lower fibrin content has been disputed by a recent meta-analysis [106]. Further histopathological studies are necessary for clarity. Higher rates of oral antithrombotic treatment among AF patients in some studies in this meta-analysis [23,53,61] may explain the elevated odds of sICH [34]. AF cohorts also reported consistently higher mean ages and baseline NIHSS scores (Table 1), which are independent predictors of poor prognosis [107–109]. There is conflicting evidence regarding how baseline blood pressure at the time of thrombolysis influences outcomes [110,111]. This is particularly relevant in the context of AF, as hypertension is more common in AF patients [112]. Furthermore, alteplase was the thrombolytic drug of choice for all IVT studies in this meta-analysis. With evolving interest in tenecteplase for AIS [113–115], future research must explore AF's role in clinical outcomes following tenecteplase.

In contrast to IVT, our analysis did not find statistically significant associations between AF and clinical outcomes following EVT. This contrasts with a meta-analysis which reported that AF was associated with significantly lower odds of favourable 90-day functional outcomes and higher odds of 90-day mortality following EVT [116]. A major difference between our analyses is that we included two additional studies [25,73]. Both studies reported no significant associations and constituted considerable weights in the meta-analyses (Figure 5). There was an outlier in the analysis for 90-day functional outcomes with a strikingly higher odds ratio [72] (Figure 5). This single-centre study contained a relatively small sample size of 83 patients and the authors acknowledged potential biases arising from the devices used for EVT varying between different clinicians. If this study is omitted from the meta-analysis, the upper limit of the confidence interval for the odds of favourable 90-day functional outcomes falls below one (Supplemental Figure S2), representing a statistically significant association. Overall, our findings suggest that the association between AF and clinical outcomes following EVT is yet to be fully understood. The underlying pathophysiological basis for potential associations also remains unresolved. Whilst non-AF patients with AIS exhibit a higher prevalence of intracranial atherosclerosis, which is associated with a refractory response to EVT [117], those with AF are more likely to have a poor collateral status [118]. Similar to the IVT studies, potential confounding arises from higher rates of comorbidities [116] and higher mean ages and baseline NIHSS scores in AF cohorts (Table 1). The varying rates of prior IVT use among the EVT cohorts could have further skewed results [27,68]. After adjusting for covariates, a meta-analysis of six randomised controlled trials (RCTs) revealed no association between AF and poorer clinical outcomes following EVT [78]. In the context of our study, we reported comorbidity rates (Table 2) and compared mean ages and baseline NIHSS scores between AF and non-AF cohorts (Table 1). However, the limitation lies in our inability to fully adjust for these covariates.

Whilst there was not a sufficient number of studies to facilitate a robust meta-analysis concerning AF and clinical outcomes following BT, the data from individual studies is presented in Table 3. The impact of BT on post-stroke outcomes is an increasingly explored field [29,30,82–84,119], with a growing interest in understanding how AF modifies this effect. However, there remains a need for more primary studies that specifically investigate

the interaction between AF and the effects of BT. Some studies have delved into potential differences in clinical outcomes between AF patients treated with BT and those receiving EVT alone [28,73,74,81,120,121]. It has been postulated that the additional IVT could be detrimental in AF patients, as it may increase the odds of sICH without enhancing functional outcomes [28]. However, other studies present contrasting findings [73,74,121], warranting a meta-analysis that pools this data for a more comprehensive analysis. This represents a crucial avenue for future research, as it strives to ascertain whether BT offers a net advantage for AF patients or if treatment might be judiciously withheld under certain circumstances.

## 5. Limitations

This study had several limitations. Firstly, the strict inclusion criteria caused numerous studies to be omitted, even though including these studies would have enabled more comprehensive meta-analyses for AF prevalence. Most prominently, the age requirement of over 18 years with no upper limit resulted in studies with a pre-defined age range (e.g., 18 to 80 years old) being excluded. Whilst the Modified Jadad Analysis (Supplemental Table S3) suggested the RCTs were generally of higher study quality, AF was usually not a focus of these trials. Thus, the data comparing clinical outcomes between AF and non-AF patients was primarily derived from cohort studies, which lack blinding and placebo controls. A large proportion of these studies were retrospective, which are further prone to selection bias and recall bias [122]. To account for this, subgroup analyses were conducted that segregated the data from prospective and retrospective studies (Supplemental Figures S3 and S4).

There was sizeable heterogeneity within numerous analyses in this study. A major contributor to this heterogeneity is AF detection methods varying from solely checking a patient's previous medical records [25,33], conducting an electrocardiogram (ECG) on admission [68], 24-h ECG monitoring [23,74] or 30-day cardiac monitoring post-discharge [71]. However, most studies did not describe how AF was detected. These variations are likely to distort the reported AF prevalence, as longer-term cardiac monitoring leads to markedly higher detection rates [123–126]. Additionally, AF is a clinically diverse condition with subtypes such as paroxysmal, persistent, and permanent AF [127,128], which most studies did not differentiate between. Limited reporting on the existing anticoagulant treatment regimen of patients also contributes to variability in the results since this influences both the risk of stroke [129] and the clinical outcomes following reperfusion therapy [130]. Due to the already expansive scope of this study, we were unable to investigate other important clinical outcomes such as haemorrhagic transformation and successful reperfusion rates. AF is also a risk factor for stroke recurrence [131], which was rarely reported upon by the studies within this meta-analysis. Lastly, whilst functional outcomes and mortality data were frequently available at the chosen 90-day timepoint, more studies are needed that investigate how AF impacts clinical outcomes at follow-up periods of one year and beyond.

## 6. Conclusions

In conclusion, our study is the first meta-analysis to our knowledge that estimates the pooled prevalence of AF amongst patients receiving reperfusion therapy following AIS. The prevalence of AF was highest in patients receiving EVT, followed by BT and IVT, respectively. AF was associated with significantly lower odds of favourable 90-day functional outcomes and significantly higher odds of sICH and 90-day mortality following IVT. Nevertheless, no significant associations between AF and these clinical outcomes were observed after EVT. While the treatment of AF using IVT and BT was examined, the limited number of available studies prevented a comprehensive meta-analysis for BT. Nevertheless, the findings of this study suggest that the treatment of AF using IVT (as assessed through current meta-analysis) and BT (based on trends from available studies [28,73,81]) is associated with poorer outcomes in contrast to EVT alone. These findings suggest potential benefits for patients diagnosed with AF, encouraging consideration of referral for EVT in

AIS management. Importantly, it must be underscored that in cases where EVT is not a feasible option, choosing IVT alone presents as a more favourable alternative than receiving no treatment at all. Awareness of these associations among clinicians is vital for informed risk stratification and effectively communicating anticipated prognoses to patients and their families. Further primary research investigating the impact of AF on clinical outcomes after BT is highly recommended.

**Supplementary Materials:** The following supporting information can be downloaded at https://www.mdpi.com/article/10.3390/neurolint15030065/s1: Search Strategy; Supplemental Table S1. Preferred Reporting Items for Systematic Reviews and Meta-Analyses (PRISMA) 2020 checklist; Supplemental Table S2. Meta-analysis of Observational Studies in Epidemiology (MOOSE) checklist.; Supplemental Table S3. Methodological quality assessment of included studies using the modified Jadad scale and assessment of funding bias.; Supplemental Table S4. Outputs from Egger's test for publication bias; Supplemental Figure S1. Forest plots of the pooled prevalence of atrial fibrillation in acute ischaemic stroke patients treated with reperfusion therapy, stratified by study type; Supplemental Figure S2. Forest plots of the association between atrial fibrillation and outcomes following intravenous thrombolysis, stratified by study type; Supplemental Figure S3. Forest plots of the association between atrial fibrillation and outcomes following endovascular thrombectomy, stratified by study type; Supplemental Figure S4. Graphs of Egger's regression tests for the meta-analyses on the association between atrial fibrillation and clinical outcomes following reperfusion therapy; Supplemental Figure S5. Funnel plots of meta-analyses on the association between atrial fibrillation and clinical outcomes following reperfusion therapy; Supplemental Figure S6. Sensitivity analyses for meta-analyses on the association between atrial fibrillation and clinical outcomes following reperfusion therapy.

**Author Contributions:** S.M.M.B. conceived the study and contributed to the planning, drafting, and revision of the manuscript and supervision of the student. S.M.M.B. encouraged J.P. to investigate and supervised the findings of this work. J.P. and S.M.M.B. wrote the first draft of this paper. All authors contributed to the revision of the manuscript. All authors have read and agreed to the published version of the manuscript.

**Funding:** This research study received no funding.

**Institutional Review Board Statement:** Not applicable. All analyses were based on previously published studies; thus, no ethical approval or patient consent was required.

**Informed Consent Statement:** Not applicable.

**Data Availability Statement:** The original contributions presented in the study are included in the article and Online Supplemental Information, and further inquiries can be directed to the corresponding author.

**Acknowledgments:** We acknowledge the financial support received from the Grant-in-Aid for Scientific Research (KAKENHI) (PI: SMMB) by the Japan Society for the Promotion of Science (JSPS), Japanese Ministry of Education, Culture, Sports, Science and Technology (MEXT). We also extend our gratitude for the JSPS International Fellowship, supported by MEXT and the Australian Academy of Science, awarded to S.M.M.B. for the period 2023–2025. The funding body has no role in the study design, data collection, analysis, interpretation of findings, or manuscript preparation. The content is solely the responsibility of the authors and does not necessarily represent the official views of the affiliated/funding organisations.

**Conflicts of Interest:** The authors declare that they have no conflict of interest.

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
