# Peer review of "Atrial Fibrillation and Reperfusion Therapy in Acute Ischaemic Stroke Patients: Prevalence and Outcomes—A Comprehensive Systematic Review and Meta-Analysis"

_2035-8377, doi:10.3390/neurolint15030065_

Round 1
Reviewer 1 Report
This meta-analysis is well written and brings new knowledge about atrial fibrillation. however, there is no current research in this area in the study presented by the authors. In my opinion, it is worth emphasizing that patients with atrial fibrillation, despite regular use of oral anticoagulants or maintaining an appropriate INR, are still exposed to a severe course of ischemic stroke because these drugs, apart from acting on the embolic mechanism, do not have a wider spectrum of effects on other risk factors. see Wankowicz et al, Bao et al.
Author Response
We sincerely appreciate your insightful feedback and the time you've dedicated to reviewing our manuscript. Your comments have greatly contributed to enhancing the quality and comprehensiveness of our study.
We have incorporated your suggestions into the "Introduction" section of the manuscript. Specifically, in lines 73-83, we have provided a more detailed explanation regarding the limitations of current anticoagulant therapies in addressing the broader spectrum of risk factors associated with atrial fibrillation (AF). We have also added additional references to relevant studies to support this discussion.
Introduction" [Lines 73-83]
"Diagnosis of AF can enable interventions such as prophylactic anticoagulant therapy, often utilizing agents like Vitamin K Antagonists (VKAs) or Non-Vitamin K Oral Anti-coagulants (NOACs), potentially preventing a considerable proportion of strokes10, 11. Notably, the chronic administration of oral anticoagulants has been shown to significantly reduce the risk of ischemic stroke by up to 64%12. Despite the consistent use of these agents or the maintenance of an appropriate International Normalized Ratio (INR), individuals with AF remain susceptible to the potential severity of ischemic stroke due to the presence of concurrent factors that frequently accompany AF13-16. It is important to acknowledge that the efficacy of these medications lies primarily in preventing embolic events, and they may not comprehensively address a broader spectrum of associated risk factors16. "
References:
10 Gladstone DJ, Bui E, Fang J, Laupacis A, Lindsay MP, Tu JV, et al. Potentially preventable strokes in high-risk patients with atrial fibrillation who are not adequately anticoagulated. Stroke. 2009; 40: 235-40.
11 Carnicelli AP, Hong H, Connolly SJ, Eikelboom J, Giugliano RP, Morrow DA, et al. Direct Oral Anticoagulants Versus Warfarin in Patients With Atrial Fibrillation: Patient-Level Network Meta-Analyses of Randomized Clinical Trials With Interaction Testing by Age and Sex. Circulation. 2022; 145: 242-55.
12 Hart RG, Pearce LA, Aguilar MI. Meta-analysis: antithrombotic therapy to prevent stroke in patients who have nonvalvular atrial fibrillation. Ann Intern Med. 2007; 146: 857-67.
13 Tokunaga K, Koga M, Itabashi R, Yamagami H, Todo K, Yoshimura S, et al. Prior Anticoagulation and Short- or Long-Term Clinical Outcomes in Ischemic Stroke or Transient Ischemic Attack Patients With Nonvalvular Atrial Fibrillation. J Am Heart Assoc. 2019; 8: e010593.
14 Evans A, Perez I, Yu G, Kalra L. Should stroke subtype influence anticoagulation decisions to prevent recurrence in stroke patients with atrial fibrillation? Stroke. 2001; 32: 2828-32.
15 Anderson JL, Horne BD, Stevens SM, Grove AS, Barton S, Nicholas ZP, et al. Randomized trial of genotype-guided versus standard warfarin dosing in patients initiating oral anticoagulation. Circulation. 2007; 116: 2563-70.
16 Wańkowicz P, Staszewski J, Dębiec A, Nowakowska-Kotas M, Szylińska A, Rotter I. Ischemic Stroke Risk Factors in Patients with Atrial Fibrillation Treated with New Oral Anticoagulants. Journal of Clinical Medicine. Vol. 10. 2021.
Your insightful suggestion regarding the potential severity of ischemic stroke in AF patients despite regular anticoagulant use aligns well with our study's context. We acknowledge the importance of recognizing that the efficacy of these medications primarily focuses on preventing embolic events, and their scope may not fully address the broader range of associated risk factors.
Thank you again for your valuable input, which has undoubtedly enriched the content and impact of our manuscript. We sincerely hope that these revisions will meet your expectations and further contribute to the understanding of managing AF-related strokes.
Sincerely,
Authors

Reviewer 2 Report
The manuscript is very well written and appears to be clear. It is a subject of interest for readers and adds clue to the management of a very prevalent disease such as atrial fibrillation. I have very few minor concerns for authors.
In the abstract there is a repetition that may be avoided (line 21-23 and 25-26).
The main results that may be revealed from the study is that AF when treated with IVT and BT is associated to a poorer outcome than EVT alone. So, it may be reasonable to believe that patients with a diagnosis of AF may be referred to EVT for AIS. The subject has been discussed by authors, but it may be added a reference and an assessment to stress that, when EVT is not possible, IVT alone is a better choice than no treatment.
Author Response
We greatly appreciate your thorough review of our manuscript. Your feedback has been instrumental in improving the quality and clarity of our work.
We have carefully addressed the concerns you raised. Specifically, we have revised the Abstract to eliminate the repetitive sentence, which has enhanced the overall clarity of the section. Additionally, we have rephrased the conclusion of the Abstract to provide a more succinct and clear summary of our key findings.
Furthermore, we have included the following paragraph based on the suggestions you made in the Conclusion (Lines 555-563).
While the treatment of AF using IVT and BT was examined, the limited number of available studies prevented a comprehensive meta-analysis for BT. Nevertheless, the findings of this study suggest that the treatment of AF using IVT (as assessed through current meta-analysis) and BT (based on trends from available studies28, 73, 81) is associated with poorer outcomes in contrast to EVT alone. These findings suggest potential benefits for patients diagnosed with AF, encouraging consideration of referral for EVT in AIS management. Importantly, it must be underscored that in cases where EVT is not a feasible option, choosing IVT alone presents as a more favorable alternative than receiving no treatment at all.
We believe that these revisions have significantly strengthened the manuscript, enhancing its clinical implications and contributing to a more comprehensive and informative discussion. We would like to express our gratitude for your thoughtful insights and constructive feedback, which have undoubtedly enhanced the overall quality of our work.
Thank you again for your time and attention to our manuscript. We hope that the revised version (please see attachment with track changes in red) meets your expectations.
Best regards,
Authors
